# `InfAlign`: Inference-aware language model alignment

**Ananth Balashankar** [* 1]   **Ziteng Sun** [* 2]   **Jonathan Berant** [1]   **Jacob Eisenstein** [1]
**Michael Collins** [1]   **Adrian Hutter** [1]   **Jong Lee** [1]   **Chirag Nagpal** [2]   **Flavien Prost** [1]   **Aradhana Sinha** [1]
**Ananda Theertha Suresh** [2]   **Ahmad Beirami** [1]

## Abstract

Language model alignment is a critical step in training modern generative language models. Alignment targets to improve win rate of a sample from the aligned model against the base model. Today, we are increasingly using inference-time algorithms (e.g., Best-of-$N$, controlled decoding, tree search) to decode from language models rather than standard sampling. We show that this train/test mismatch makes standard RL framework sub-optimal in view of such inference-time methods. To this end, we propose a framework for inference-aware alignment (`InfAlign`), which aims to optimize *inference-time win rate* of the aligned policy against the base model. We prove that for any inference-time decoding procedure, the optimal aligned policy is the solution to the standard RLHF problem with a *transformation of the reward*. This motivates us to provide the calibrate-and-transform RL (`InfAlign-CTRL`) algorithm to solve this problem, which involves a reward calibration step and a KL-regularized reward maximization step with a transformation of the calibrated reward. For best-of-$N$ sampling and best-of-$N$ jailbreaking, we propose specific transformations offering up to 3-8% improvement on inference-time win rates. Finally, we also show that our proposed reward calibration method is a strong baseline for optimizing standard win rate.

---

[*]Equal contribution   [1]Google DeepMind [2]Google Research.
Correspondence to:
Ananth Balashankar <ananthbshankar@google.com>,
Ziteng Sun <zitengsun@google.com>,
Jonathan Berant <joberant@google.com>,
Jacob Eisenstein <jeisenstein@google.com>,
Ananda Theertha Suresh <theertha@google.com>,
Ahmad Beirami <ahmad.beirami@gmail.com>.

*Proceedings of the 42nd International Conference on Machine Learning*, Vancouver, Canada. PMLR 267, 2025. Copyright 2025 by the author(s).

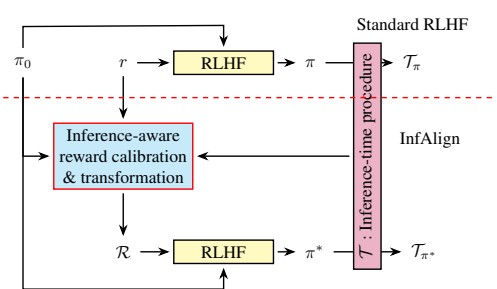

*Figure 1.* Given an inference-time procedure such as Best-of-$N$, standard RLHF suffers from a train/test mismatch between train-time policy $\pi$ and inference-time policy $\mathcal{T}_\pi$. `InfAlign` bridges the gap by optimizing a policy-transformed reward $\mathcal{R}$, yielding a policy, $\pi^*$, that is optimized for inference under $\mathcal{T}_{\pi^*}$.

## 1. Introduction

Aligning language models (LMs) through reinforcement learning from human feedback (RLHF) is a widely adopted finetuning framework to improve a reward (e.g., safety or quality). RLHF generally entails training a reward model, and then solving a KL-regularized RL problem (Christiano et al., 2017; Stiennon et al., 2020; Ouyang et al., 2022). Other ways of solving RLHF include variants of direct preference optimization (Rafailov et al., 2023; Azar et al., 2023), reward model distillation (Fisch et al., 2024), and Best-of-$N$ distillation (Gui et al., 2024; Amini et al., 2024; Sessa et al., 2024). The success of RLHF is typically measured through the win rate of the aligned model against the base model, i.e., how often a sample from the aligned model wins against the base model using a judge for the task.

However, rarely is the aligned model used as is at inference time; instead an *inference-time procedure* is typically used to accomplish a task. For example, it is customary to perform one or more of the following procedures at decoding: best-of-$N$ sampling (Nakano et al., 2022; Beirami et al., 2024), best-of-$N$ jailbreaking (Hughes et al., 2024b), chain-of-thought reasoning (Wei et al., 2022; OpenAI, 2024), and self consistency (Wang et al., 2022a; DeepSeek-AI, 2025) (see Appendix A for a more comprehensive discussion).

In this paper, we address the following question: *Can we better align a language model to be served with a known inference-time procedure?* We propose a family of optimization objectives, which we call the *inference-aware alignment*

(`InfAlign`) framework. The objective optimizes for the *inference-time win rate* against the base model with KL regularization, where inference-time win rate entails obtaining a response from each model through the *inference-time procedure* and counting which sample wins. While directly optimizing the inference-time win rate seems intractable, we prove that the solution could be obtained by solving RLHF with a specific transformation of the reward (Lemma 1). Therefore, the challenge of optimizing for inference-time win rate can be captured by designing a reward transformation that is suited to the specific inference-time procedure. We present a diagram of `InfAlign` in Fig. 1.

We show that the optimal reward transformation satisfies a coupled-transformed reward/policy optimization objective which lends itself to iterative optimization for a large class of inference-time procedures (Theorem 1). However, the approach is unfortunately computationally inefficient and infeasible for real-world models.

To enable practical solutions to `InfAlign`, we propose Calibrate-and-Transform Reinforcement Learning (`InfAlign-CTRL`), which adopts a three-step approach: (1) *calibrate* the scores of the reward model with respect to responses sampled per-prompt by the reference model; (2) *transform* the calibrated scores for a given inference-time procedure; and (3) solve the RLHF problem with the transformed reward. We prove various desirable properties of the reward calibration step and empirically show that it reduces reward hacking and improves standard win rate. Reward transformation allows us to further tailor the objective based on the inference-time procedure. Different choices of transformation lead to popular existing alignment objectives such as IPO (Azar et al., 2023) and best-of-$N$ distillation (Gui et al., 2024; Amini et al., 2024; Sessa et al., 2024).

We particularize the study to two simple yet popular inference-time strategies: Best-of-$N$ (`BoN`) sampling (Nakano et al., 2022; Beirami et al., 2024) and Best-of-$N$ jailbreaking (Hughes et al., 2024a), which we call Worst-of-$N$ (`WoN`) since the defender may assume that the attacker is choosing the worst outcome of $N$ samples. Despite simplicity, `BoN` is known to be an effective procedure for inference-time alignment (Beirami et al., 2024; Gui et al., 2024; Mudgal et al., 2024) and is the dominant approach in scaling inference-time compute (Snell et al., 2024; Brown et al., 2024). Variants of `WoN` are effective and popular for evaluating safety against jailbreaks (Yohsua et al., 2024; Chao et al., 2023; Souly et al., 2024; Hughes et al., 2024b; Beetham et al., 2024; Mehrabi et al., 2023). We find suitable reward transformations for these procedures through the `InfAlign` framework.

Empirically, we apply `InfAlign-CTRL` to the Anthropic helpfulness (Bai et al., 2022), and Reddit summarization dataset (Stiennon et al., 2020) for optimizing `BoN` perfor-

mance and Anthropic harmlessness for optimizing `WoN` performance with various values of $N$. We show that solving `InfAlign` through `InfAlign-CTRL` outperforms various SOTA RLHF solvers at inference-time win rate by 3-8%. We also show `InfAlign-CTRL` (with no reward transformation) is on-par with SOTA methods for optimizing the standard win rate, with improved win rate for Anthropic helpfulness and harmlessness tasks.

**Organization.** In Section 2, we provide the `InfAlign` problem setup. In Section 3, we introduce RLHF with reward transformation as the main framework to solve `InfAlign`. In Section 4, we present the `InfAlign-CTRL` method, discuss properties of calibration, and present practical implementations. In Section 5, we provide experimental results.

## 2. Problem setup

We consider a generative language model that produces a response conditioned on an input prompt. Given a prompt $\boldsymbol{x}$, e.g., $\boldsymbol{x} = $ *What is a large language model?*, a generative language model $\pi$, or a policy, specifies a conditional distribution $\pi(\cdot \mid \boldsymbol{x})$, from which the response $\boldsymbol{y}$ should be generated. We use $\mathcal{X}$ and $\mathcal{Y}$ to denote the space of possible inputs and outputs, respectively. Throughout the paper, we assume $\pi_{\mathrm{ref}}$ is a fixed base policy, e.g., obtained from supervised finetuning. We will often use the notation $\pi(\boldsymbol{y} \mid \boldsymbol{x}) \propto f(\boldsymbol{y})$ for some $f : \mathcal{Y} \to \mathbb{R}_+$ to denote the conditional distribution $\pi(\boldsymbol{y} \mid \boldsymbol{x}) = f(\boldsymbol{y})/(\sum_{\boldsymbol{y}} f(\boldsymbol{y}))$ obtained after normalization.

**Alignment of language models.** Let $r : \mathcal{X} \times \mathcal{Y} \to \mathbb{R}$ be a reward function that assigns a scalar value to any (prompt, response) pair, e.g., a model trained side-by-side on human preferences. The reward determines the goodness of response $\boldsymbol{y}$ in context $\boldsymbol{x}$. The goal of language model alignment is to construct an aligned distribution $\pi$ that improves the reward of the response while being close to $\pi_{\mathrm{ref}}$. We introduce the popular KL-regularized reinforcement learning (RL) framework, also known as RLHF.

**Definition 1** (KL-regularized RL). *Let $\beta > 0$ be a regularization parameter, the KL-regularized RL (RLHF) problem aims to maximize the expected reward with a KL regularizer below:*[1]
$$\pi_{r,\beta}^*(\cdot \mid \boldsymbol{x}) = \arg \max_{\pi} \left\{ \mathcal{L}_{\pi_{\mathrm{ref}}, r, \beta} \right\},$$

*where*
$$\mathcal{L}_{\pi_{\mathrm{ref}}, r, \beta} = \mathbb{E}_{\boldsymbol{y} \sim \pi(\cdot \mid \boldsymbol{x})} \{ r(\boldsymbol{x}, \boldsymbol{y}) \} - \beta D_{KL}(\pi(\cdot \mid \boldsymbol{x}) \| \pi_{\mathrm{ref}}(\cdot \mid \boldsymbol{x})).$$

When evaluating an aligned policy, a common measure to use is the win rate (Stiennon et al., 2020; Hilton & Gao,

---

[1]The solution to this optimization problem is unique and admits a closed-form expression (Korbak et al., 2022; Rafailov et al., 2023; Yang et al., 2024).

2022) over the base policy $\pi_{\text{ref}}$. Let

$$w_r(\boldsymbol{y}, \boldsymbol{z}|\boldsymbol{x}) := \mathbb{1}\{r(\boldsymbol{x}, \boldsymbol{y}) > r(\boldsymbol{x}, \boldsymbol{z})\} + \frac{\mathbb{1}\{r(\boldsymbol{x}, \boldsymbol{y}) = r(\boldsymbol{x}, \boldsymbol{z})\}}{2}$$

be the win random variable under reward $r$. We define the calibrated reward and win rate below.

**Definition 2** (Calibrated reward). *The calibrated reward* [2] $\mathcal{C}_{r,\pi}(\boldsymbol{x}, \boldsymbol{y})$ *under policy $\pi$ is defined below*

$$\mathcal{C}_{r,\pi}(\boldsymbol{x}, \boldsymbol{y}) := \mathbb{E}_{\boldsymbol{z} \sim \pi(\cdot|\boldsymbol{x})}\{w_r(\boldsymbol{y}, \boldsymbol{z} \mid \boldsymbol{x})\}. \quad (1)$$

**Definition 3** (Standard win rate). *For any policy $\pi_1$ and $\pi_2$, the standard win rate (or win rate) of policy $\pi_1$ over policy $\pi_2$ given prompt $\boldsymbol{x}$, as measured by reward $r$ is defined as:*

$$W_r(\pi_1 \succ \pi_2 \mid \boldsymbol{x}) := \mathbb{E}_{\boldsymbol{y} \sim \pi_1(\cdot|\boldsymbol{x}), \boldsymbol{z} \sim \pi_2(\cdot|\boldsymbol{x})}\{w_r(\boldsymbol{y}, \boldsymbol{z} \mid \boldsymbol{x})\}.$$

Moreover, it can be shown that $W_r(\pi_1 \succ \pi_2|\boldsymbol{x}) = \mathbb{E}_{\boldsymbol{y} \sim \pi_1(\cdot|\boldsymbol{x})}\{\mathcal{C}_{r,\pi_2}(\boldsymbol{x}, \boldsymbol{y})\}$.

**Inference-time procedure** ($\mathcal{T}$). As mentioned earlier, in many cases, decoding is done through an inference-time procedure. The obtained sample follows a transformed distribution that depends on the policy and the inference-time procedure. Let $\Delta_{\mathcal{Y}}$ be the set of possible distributions over the output set $\mathcal{Y}$. In this work, we model an inference-time procedure as a mapping between distributions over the output space $\mathcal{T} : \Delta_{\mathcal{Y}} \to \Delta_{\mathcal{Y}}$,

$$\pi(\cdot \mid \boldsymbol{x}) \xrightarrow{\mathcal{T}} \mathcal{T}_{\pi}(\cdot \mid \boldsymbol{x}),$$

where $\mathcal{T}_{\pi}(\cdot \mid \boldsymbol{x})$ denotes the distribution of responses conditioned on $\boldsymbol{x}$ after the inference-time procedures is applied. In these cases, it is customary to compare the models by considering the following inference-time win rate of the aligned policy $\pi$.

**Definition 4** (Inference-time win rate). *Under inference-time processing $\mathcal{T}$, the inference-time win rate of policy $\pi_1$ over $\pi_2$ is defined as*

$$W_r^{\mathcal{T}}(\pi_1 \succ \pi_2|\boldsymbol{x}) := \mathbb{E}_{\boldsymbol{y} \sim \mathcal{T}_{\pi_1}(\cdot|\boldsymbol{x}), \boldsymbol{z} \sim \mathcal{T}_{\pi_2}(\cdot|\boldsymbol{x})}\{w_r(\boldsymbol{y}, \boldsymbol{z}|\boldsymbol{x})\}.$$

Similarly, it can be shown that

$$W_r^{\mathcal{T}}(\pi_1 \succ \pi_2 \mid \boldsymbol{x}) = \sum_{\boldsymbol{y}} \mathcal{C}_{r,\mathcal{T}_{\pi_2}}(\boldsymbol{x}, \boldsymbol{y})\mathcal{T}_{\pi_1}(\boldsymbol{y} \mid \boldsymbol{x}), \quad (2)$$

where $\mathcal{C}_{r,\mathcal{T}_{\pi_2}}(\boldsymbol{x}, \boldsymbol{y})$ is the calibrated reward under the inference-time policy $\mathcal{T}_{\pi_2}$ (see Proof in Appendix B.2).

With the above definitions at hand, the goal of InfAlign is solve the following KL-regularized inference-time win rate maximization problem.

**Definition 5** (InfAlign). *For a given inference-time procedure $\mathcal{T}$ and $\beta > 0$, InfAlign solves the following KL-regularized inference-time win rate maxmization problem:*

$$\max_{\pi}\{W_r^{\mathcal{T}}(\pi \succ \pi_{\text{ref}}|\boldsymbol{x}) - \beta D_{KL}(\pi(\cdot|\boldsymbol{x})\|\pi_{\text{ref}}(\cdot|\boldsymbol{x}))\}. \quad (3)$$

The formulation of optimizing standard win-rate vs KL tradeoff has been previous studied for RLHF in Gui et al. (2024); Azar et al. (2023). The above formulation reduces to the IPO objective (Azar et al., 2023, Equation (8)) when $\mathcal{T}$ is the identity transformation and extends it otherwise for an arbitrary inference-time procedure.[3] In practice, the win rate is often evaluated by a judge different from the training time reward $r$. In this paper, we stick to $r$ when developing the algorithms as in standard RLHF framework (Definition 1). In experiments, we use a separate judge from the reward model.

**Continuous language models.** For simplicity of the presentation and analysis, some of the theoretical results will be based on the assumption that $\mathcal{Y}$ is a continuous set, and the language model has a density over $\mathcal{Y}$. We also assume that $r$ assigns distinct rewards to different $\boldsymbol{y}$'s for a given $\boldsymbol{x}$. Note that we don't make any such assumptions when providing our algorithmic developments, and the experimental results.

These assumptions have been made in the past implicitly by (Hilton & Gao, 2022) to estimate the KL divergence of Best-of-$n$ policy, and by (Gui et al., 2024) to characterize the KL divergence and win rate tradeoffs. While these assumptions lead to approximations when analyzing real-world distributions, the results derived under them are reasonably tight when the actual likelihood of the language model outcomes are small (Beirami et al., 2024).

## 3. Reinforcement learning with reward transformation

In this section, we propose a general framework for solving InfAlign (Definition 5). Our approach is based on designing a new reward function $\mathcal{R}$ based on the reward model $r$, the inference-time procedure $\mathcal{T}$, and the base policy $\pi_{\text{ref}}$, such that solving the RLHF problem (Definition 1) with the transformed reward $\mathcal{R}$ leads to an optimal solution to InfAlign. More precisely, the aligned policy is the maximizer of the following regularized objective:

$$\mathcal{R}_{r,\pi_{\text{ref}},\mathcal{T}}(\boldsymbol{x}, \boldsymbol{y}) - \beta D_{\text{KL}}(\pi(\cdot \mid \boldsymbol{x})\|\pi_{\text{ref}}(\cdot \mid \boldsymbol{x})), \quad (4)$$

where $\mathcal{R}_{r,\pi_{\text{ref}},\mathcal{T}}(\boldsymbol{x}, \boldsymbol{y})$ is a transformed reward function. Interestingly, we show that such reward transformation is

---

[2] The definition is similar to the cumulative density function (CDF) of the reward $r(\boldsymbol{x}, \boldsymbol{y})$ under $\pi$ except for how ties are decided.

[3] One question that arises is the role of the KL divergence regularizer in Eq. (3). We argue that the regularizer essentially enables multi-tasking between the SFT task and the RL task, which we formally prove for log-linear models in Appendix C. In other words, the KL divergence regularizer enables to preserve/distill the core capabilities of the SFT model while acquiring a new one through the RLHF process.

sufficient to solve `InfAlign`.

**Lemma 1.** *For any base policy $\pi_{\mathrm{ref}}$, reward model $r$, inference-time procedure $\mathcal{T}$, and $\beta > 0$, there exists a reward function $\mathcal{R}_{r,\pi_{\mathrm{ref}},\mathcal{T}}$ such that the maximizer of Eq.* (4) *solves the optimization problem in Eq.* (3) *(Definition 5).*

In general, such optimal reward transformation will depend on the base policy $\pi_{\mathrm{ref}}$, the inference-time procedure $\mathcal{T}$, and the reward model $r$. In the lemma below, we list the property that the reward transformation and the resulting optimal aligned policy must satisfy.

**Theorem 1** (Characterization of `InfAlign` solution). *Assuming that $\mathcal{T}$ is such that $\partial \mathcal{T}_\pi(\boldsymbol{y}_1 \mid \boldsymbol{x})/\partial \pi(\boldsymbol{y}_2 \mid \boldsymbol{x})$[4] exists for all $\boldsymbol{x}, \boldsymbol{y}_1, \boldsymbol{y}_2$, then we have the optimal transformed reward $\mathcal{R}$ and the optimal policy $\pi^*$ in Eq.* (3) *must satisfy the following coupled equations: $\forall \boldsymbol{x}, \boldsymbol{y}$*

$$\pi^*(\boldsymbol{y}|\boldsymbol{x}) \propto \pi_{\mathrm{ref}}(\boldsymbol{y} \mid \boldsymbol{x})e^{\frac{1}{\beta}\mathcal{R}(\boldsymbol{x},\boldsymbol{y})} \quad (5)$$

$$\mathcal{R}(\boldsymbol{x},\boldsymbol{y}) = \frac{\partial}{\partial \pi(\boldsymbol{y} \mid \boldsymbol{x})}W_r^{\mathcal{T}}(\pi \succ \pi_{\mathrm{ref}} \mid \boldsymbol{x})|_{\pi=\pi^*} \quad (6)$$

$$= \sum_{\boldsymbol{z}} \mathcal{C}_{r,\mathcal{T}_{\pi_{\mathrm{ref}}}}(\boldsymbol{x},\boldsymbol{z})\frac{\partial \mathcal{T}_\pi(\boldsymbol{z} \mid \boldsymbol{x})}{\partial \pi(\boldsymbol{y} \mid \boldsymbol{x})}|_{\pi=\pi^*}, \quad (7)$$

*where the last inequality is due to Eq.* (2).

Missing proofs are presented in Appendix B. Theorem 1 naturally leads to an iterative EM-style algorithm that (I) updates $\pi$ with $\mathcal{R}$ fixed based on Eq. (5) and (II) updates $\mathcal{R}$ with $\pi$ fixed based on Eq. (7) until convergence. However, such algorithm suffers from two drawbacks: first, for general language models, it is inefficient/intractable to evaluate Eq. (7) since it involves evaluating the policy on a large, or even infinite output space; second, it is unclear whether such an algorithm could lead to the optimal solution.

To find more efficient ways to design reward transformations, we examine the case when no inference-time procedure is performed. In this case, $\mathcal{T}_\pi = \pi$ and

$$\frac{\partial}{\partial \pi(\boldsymbol{y} \mid \boldsymbol{x})}\mathcal{T}_\pi(\boldsymbol{z} \mid \boldsymbol{x}) = \mathbb{1}\{\boldsymbol{z} = \boldsymbol{y}\}.$$

Eq. (7) will reduce to $\mathcal{R}(\boldsymbol{x},\boldsymbol{y}) = \mathcal{C}_{r,\pi_{\mathrm{ref}}}(\boldsymbol{x},\boldsymbol{y})$, the calibrated reward under $\pi_{\mathrm{ref}}$.

**Corollary 1.** *When no inference-time procedure is performed, i.e. $\forall \pi, \mathcal{T}_\pi = \pi$, the maximizer of Eq.* (4) *with $\mathcal{R}(\boldsymbol{x},\boldsymbol{y}) = \mathcal{C}_{r,\pi_{\mathrm{ref}}}(\boldsymbol{x},\boldsymbol{y})$ is the solution to Eq.* (3).

The above corollary is also observed in Azar et al. (2023); Gui et al. (2024). Hence Theorem 1 can be viewed as a

---

[4]To make sure that the partial derivative is well-defined, we assume that $\mathcal{T}$ is well-defined for inputs within an infinitesimal expansion of $\Delta_{\boldsymbol{y}}$. We note that the assumption holds for procedures like `BoN` and `WoN`, as shown in Lemma 5.

generalization of these results with general inference-time procedures. The observation motivates us to consider RLHF with a specific family of reward transformations that involves a reward calibration step, described next.

## 4. `InfAlign-CTRL`: Calibrate-and-transform reinforcement learning

In this section, we propose the `InfAlign-CTRL` method, which is our proposed solver for the `InfAlign` problem. The method consists of: (1) *Calibration*: Approximate the calibrated reward $\mathcal{C}_{r,\pi_{\mathrm{ref}}}$; (2) *Transformation*: Apply transformation $\Phi$ on top of $\mathcal{C}_{r,\pi_{\mathrm{ref}}}$ to obtain $\mathcal{R}_\Phi = \Phi \circ \mathcal{C}_{r,\pi_{\mathrm{ref}}}$; (3) *Solve RLHF with the transformed reward*. We discuss each step in more details below.

**Reward calibration.** The goal of this step is to obtain the calibrated reward $\mathcal{C}_{r,\pi_{\mathrm{ref}}}$ (Definition 2). We first assume $\mathcal{C}_{r,\pi_{\mathrm{ref}}}$ could be obtained perfectly and discuss practical ways to approximate it in Section 4.4. In Section 4.1, we discuss various properties of $\mathcal{C}_{r,\pi_{\mathrm{ref}}}$ In particular, it can be shown that for continuous language models, the distribution of $\mathcal{C}_{r,\pi_{\mathrm{ref}}}$ is independent from the base policy, and the reward model (Lemma 4), which provides a unified view of the outputs from a language model through the lens of calibrated reward. This allows us to focus the design of the transformation function $\Phi$ on $\mathcal{T}$ in the following step.

As mentioned in Corollary 1, using the calibrated reward directly in the RLHF problem leads to an optimal solution to `InfAlign` with no inference-time procedure. Note that the resulting solution is the same as the alignment objective of Azar et al. (2023). Moreover, it can also be shown that using $\log \mathcal{C}_{r,\pi_{\mathrm{ref}}}$ as the reward, the solution to the RLHF problem recovers the popular best-of-$N$ distillation objective, which has been studied in a recent line of works (Gui et al., 2024; Amini et al., 2024; Sessa et al., 2024), and shown to be nearly optimal for standard win rate (Yang et al., 2024; Gui et al., 2024). We note that while these methods lead to similar optimization objectives, `InfAlign-CTRL` makes this calibration step explicit, resulting in a different algorithmic approach. In Section 5, we compare the results with the abovementioned baseline approaches on standard win rate, and show that it leads to improved win rate. We also empirically show that reward calibration is beneficial to mitigate reward hacking (Appendix D).

**Reward transformation.** The goal is to further transform the calibrated reward using a transformation function $\Phi : [0,1] \to \mathbb{R}$. The function $\Phi$ is chosen based on the inference-time procedure $\mathcal{T}$ so that $\Phi \circ \mathcal{C}_{r,\pi_{\mathrm{ref}}}$ is a good reward transformation to use for solving `InfAlign`.

Ideally, we would like the design of $\Phi$ to only depend on the inference-time procedure $\mathcal{T}$ so that it is transferable among different $r$ and $\pi_{\mathrm{ref}}$. A natural question is for what

type of $\mathcal{T}$'s this is possible. In Section 4.2, we show that for *calibrated* inference-time procedures (Definition 6), and continuous language models, it is sufficient for $\Phi$ to depend on $\mathcal{T}$. And in these cases, different $\Phi$'s could be evaluated efficiently with simple language models that can be easily simulated, which enables the search for good or even optimal transformations. We use Best-of-$N$ and Worst-of-$N$ as examples of inference-time procedures to demonstrate the effectiveness of such approach in Section 4.3.

**RLHF with the transformed reward.** At this step, we solve the RLHF problem with reward function,

$$\mathcal{R}_\Phi(\boldsymbol{x}, \boldsymbol{y}) = \Phi(\mathcal{C}_{r, \pi_{\text{ref}}}(\boldsymbol{y} \mid \boldsymbol{x})),$$

to obtain the aligned policy $\pi^*_{\mathcal{R}_\Phi, \beta}$. Since we only modify the reward function, the step can be performed with standard RLHF solvers. We present a practical version of the `InfAlign-CTRL` method in Section 4.4.

## 4.1. Properties of reward calibration

We present properties of $\mathcal{C}_{r, \pi_{\text{ref}}}$. The first property states that reward calibration preserves the ordering of the reward.

**Lemma 2** (Calibration is a bounded monotone increasing transformation of reward). *We have $\mathcal{C}_{r, \pi_{\text{ref}}}(\boldsymbol{x}, \boldsymbol{y}) \in [0, 1]$. Furthermore, we have for any $\boldsymbol{y}$ and $\boldsymbol{z}$*

$$r(\boldsymbol{x}, \boldsymbol{y}) \geq r(\boldsymbol{x}, \boldsymbol{z}) \implies \mathcal{C}_{r, \pi_{\text{ref}}}(\boldsymbol{x}, \boldsymbol{y}) \geq \mathcal{C}_{r, \pi_{\text{ref}}}(\boldsymbol{x}, \boldsymbol{z}).$$

Moreover, $\mathcal{C}_{r, \pi_{\text{ref}}}$ is invariant under all monotone increasing transformations of the reward function, stated below.

**Lemma 3** (Calibration is invariant under monotone increasing transformations). *Let $m : \mathbb{R} \to \mathbb{R}$ be any strictly monotone increasing function. Then $\mathcal{C}_{m(r), \pi_{\text{ref}}} = \mathcal{C}_{r, \pi_{\text{ref}}}$.*

This property is useful since as long as the learned reward model $r$ can capture relative human preference between each pair of generations, the calibration of $r$ will remain unchanged, making $\mathcal{C}_{r, \pi_{\text{ref}}}$ more robust to the learning of $r$.

The next property shows that the calibration operation allows us to transform the distribution of the reward under the base policy to a uniform distribution over $[0, 1]$ regardless of the base policy $\pi_{\text{ref}}$ and the reward model $r$.

**Lemma 4.** *If $\pi$ is a continuous language model, let $\boldsymbol{y}$ be sampled from $\pi_{\text{ref}}(\cdot \mid \boldsymbol{x})$, then we have $\forall \boldsymbol{x}$,*

$$\mathcal{C}_{r, \pi_{\text{ref}}}(\boldsymbol{x}, \boldsymbol{y}) \sim \text{Unif}([0, 1]).$$

The lemma provides us a unified view of the output from a language model through the space of calibrated reward.

## 4.2. Reward transformation for calibrated inference-time procedure

We consider a family of inference-time procedures that only depend on the calibrated reward of the outputs, which we term *calibrated procedures*, and discuss how to design a suitable $\Phi$ for this family of transformations. We first define *calibrated procedures* below.

**Definition 6** (Calibrated inference-time procedure). *An inference-time procedure $\mathcal{T}$ is called a* calibrated procedure *if there exists a mapping function $g_\mathcal{T} : [0, 1] \to \mathbb{R}$ such that for any $\pi$, $r$, and $\boldsymbol{x}, \boldsymbol{y}$, we have*

$$\mathcal{T}_\pi(\boldsymbol{y} \mid \boldsymbol{x}) \propto \pi(\boldsymbol{y} \mid \boldsymbol{x}) \cdot g_\mathcal{T}(\mathcal{C}_{r, \pi}(\boldsymbol{x}, \boldsymbol{y})).$$

Our next result shows that for calibrated inference-time procedures and continuous language models, the aligned policy obtained from `InfAlign-CTRL` with any $\Phi$ has a win rate and KL divergence independent of $\pi_{\text{ref}}$ and $r$.

**Theorem 2** (Model-agnostic property of calibrated inference-time procedures, informal version of Theorem 4). *If $\mathcal{T}$ is a calibrated inference-time procedure, for any continuous language model $\pi$, $\beta > 0$ and transformation function $\Phi$, we have that both $W_r^\mathcal{T}(\pi^*_{\mathcal{R}_\Phi, \beta} \succ \pi_{\text{ref}} \mid \boldsymbol{x})$ and $D_{KL}(\pi^*_{\mathcal{R}_\Phi, \beta} \| \pi_{\text{ref}})$ are independent of $r$ and $\pi_{\text{ref}}$.*

The above theorem allows us to evaluate a transformation $\Phi$ by focusing on simple continuous language models that are easy to compute and simulate. In the next section, we focus on Best-of-$N$ and Worst-of-$N$, as examples to demonstrate how the theorem enables us to efficiently evaluate different transformations in practical scenarios.

## 4.3. Finding reward transformations for `BoN` and `WoN`

**Best-of-$N$ inference-time procedure (`BoN`).** During inference, $N$ i.i.d. responses from a policy $\pi$ are generated. The final output is the one with the highest reward, i.e.,

$$\boldsymbol{y}_{\text{BoN}} = \arg \max_{\boldsymbol{y} \in \{\boldsymbol{y}_1, \cdots \boldsymbol{y}_N\}} r(\boldsymbol{x}, \boldsymbol{y}).$$

**Worst-of-$N$ inference-time procedure (`WoN`).** $N$ i.i.d. responses from a policy $\pi$ are generated, and the final output is the one with the lowest reward. i.e.,

$$\boldsymbol{y}_{\text{WoN}} = \arg \min_{\boldsymbol{y} \in \{\boldsymbol{y}_1, \cdots \boldsymbol{y}_N\}} r(\boldsymbol{x}, \boldsymbol{y}).$$

The lemma below presents the distribution of outputs after the inference-time procedure is performed.

**Lemma 5.** *For any $N$ and continuous language model $\pi$,*

$$\text{BoN}_\pi(\boldsymbol{y} \mid \boldsymbol{x}) = N \cdot \pi(\boldsymbol{y} \mid \boldsymbol{x}) \cdot \mathcal{C}_{r, \pi}(\boldsymbol{x}, \boldsymbol{y})^{N-1}.$$

$$\text{WoN}_\pi(\boldsymbol{y} \mid \boldsymbol{x}) = N \cdot \pi(\boldsymbol{y} \mid \boldsymbol{x}) \cdot (1 - \mathcal{C}_{r, \pi}(\boldsymbol{x}, \boldsymbol{y}))^{N-1}.$$

Note that the results for `BoN` have already been derived previously (Beirami et al., 2024; Gui et al., 2024; Amini et al., 2024). The lemma shows that these two inference-time procedures are calibrated procedures so that as claimed

in Theorem 2, for the aligned policy, the inference-time win rate and KL divergence deviation from the base policy are independent of the base policy and reward model. Below we present the precise formula for these two procedures.

**Theorem 3** (Properties of BoN and WoN procedures). *For any transformation function $\Phi$, the solution $\pi^*_{\mathcal{R}_\Phi, \beta}$ to* `InfAlign-CTRL` *satisfies the followings: Let $F_{\Phi,\beta}(u) = \frac{\int_0^u e^{\Phi(u')/\beta} du'}{\int_0^1 e^{\Phi(u')/\beta} du'}$.*

- *For any $\boldsymbol{x}$, $W_r^{BoN}(\pi^*_{\mathcal{R}_\Phi,\beta} \succ \pi_{\text{ref}} \mid \boldsymbol{x}) =$*

$$1 - N \int_0^1 F_{\Phi,\beta}(u)^N u^{N-1} du, \quad (8)$$

- *For any $\boldsymbol{x}$, $W_r^{WoN}(\pi^*_{\mathcal{R}_\Phi,\beta} \succ \pi_{\text{ref}} \mid \boldsymbol{x}) =$*

$$N \int_0^1 \left(1 - F_{\Phi,\beta}(u)\right)^N (1 - u)^{N-1} du, \quad (9)$$

- $D_{KL}(\pi^*_{\mathcal{R}_\Phi,\beta} \| \pi_{\text{ref}}) =$

$$\frac{1}{\beta} \frac{\int_0^1 \Phi(u) e^{\Phi(u)/\beta} du}{\int_0^1 e^{\Phi(u)/\beta} du} - \log\left(\int_0^1 e^{\Phi(u)/\beta} du\right). \quad (10)$$

The above theorem generalizes the win rate calculation in Gui et al. (2024, Lemma 5) to inference-time win rate. By varying $\beta$ in Theorem 3, we obtain an alignment curve plotting the inference-time win rate and KL divergence for different aligned policies. This allows us to compare the performance of different transformation functions.

In the rest of the section, we investigate different types of transformations, and analytically compute the alignment curves, i.e., the plot of $(D_{\text{KL}}(\pi^*_{\mathcal{R}_\Phi,\beta} \| \pi_{\text{ref}}), W_r^T(\pi^*_{\mathcal{R}_\Phi,\beta} \succ \pi_{\text{ref}})$ for different $\beta$'s. The transformations we consider include optimal transformations for standard win rate, exponential functions, and optimization-based transformations.

*Optimal reward transformations for standard win rate.* The identity mapping $\Phi(x) = x$ proposed by Azar et al. (2023) and the logarithmic mapping $\Phi(x) = \log x$ as used by BoN distillation (Beirami et al., 2024; Yang et al., 2024; Gui et al., 2024; Amini et al., 2024; Sessa et al., 2024) are shown to be (almost) optimal for the standard win rate. We investigate whether these transformations are suited to inference-time procedures.

*Deriving an optimized reward transformation function.* Due to Theorem 2, one can optimize for good $\Phi$'s using simple toy language models, leading to the following corollary.

**Corollary 2.** *For any $\beta > 0$, the $\Phi$ that solves* `InfAlign` *with $\mathcal{T} = $ BoN must satisfy the following pair of equations:*

$$\Phi_{BoN}(u) = -N^2 \int_u^1 F(v)^{N-1} v^{N-1} dv,$$

$$f(u) \propto e^{\frac{\Phi_{BoN}(u)}{\beta}},$$

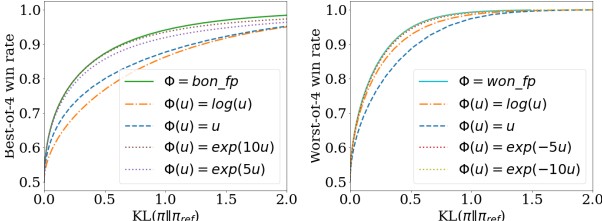

*Figure 2.* Best-of-$N$ (left) and Worst-of-$N$ (right) win rate vs KL tradeoff curves for $N = 4$ with different transformation functions.

*where $F(v) = \int_0^v f(v) dv$ is the CDF with density function $f$. and for $\mathcal{T} = $ WoN, it satisfies that*

$$\Phi_{WoN}(u) = -N^2 \int_u^1 (1-v)^{N-1} (1 - F(v))^{N-1} dv,$$

$$f(u) \propto e^{\frac{\Phi_{WoN}(u)}{\beta}}.$$

Hence, we derive a transformation function by iteratively finding the fixed point of the coupled equations in Corollary 2. We call them `bon_fp` and `won_fp`.

*Exponential tilting for reward transformation.* In addition to deriving the optimized transformation, motivated by the exponential tilting of loss functions (Li et al., 2021; 2023), we consider the following exponential transformation:

$$\Phi_t(u) = \text{sign}(t) \cdot e^{tu}, \quad (11)$$

where $\text{sign}(t) = 1$ for $t \geq 0$ and $\text{sign}(t) = -1$ for $t < 0$. These exponential transformations are essentially helping to optimize different quantiles of the reward for different values of $t$ (Li et al., 2023). For a positive value of $t$, the exponential tilting transformation focuses on optimizing the higher quantiles of the objective (calibrated reward) . On the other hand, for a negative value of $t$, the transformation is akin to optimizing the lower quantiles of the calibrated reward, which makes it a suitable transformation for the WoN inference-time procedure.

**Results.** We compare different reward transformations on inference-time win rate vs KL divergence for three inference-time procedures: {standard, BoN, WoN } with different $N$'s. The tradeoff curves are obtained by varying the strength of the regularizer, $\beta$. We present the result for $N = 4$ in Fig. 2 and provide more results in Appendix F.

For Best-of-4 win rate, the identity and log transformation, which are optimal for standard win rate, are suboptimal. The best tradeoffs are given by `bon_fp`. We also observe that $\exp(10x)$ are almost as good as `bon_fp` for Best-of-4. For Worst-of-4 win rate, it is observed that `won_fp` gives the best tradeoffs for this inference-time procedure. Here, $\exp(-5x)$ and $\exp(-10x)$ are almost as good for Worst-of-2 and Worst-of-4, respectively. We also observe that identity transformation and log transformation are sub-optimal in these cases and the log transformation gives better tradeoffs for WoN compared to the identity transformation.

The above results show that considering standard win rate as the only metric is not sufficient when inference-time procedure is concerned, demonstrating the importance of inference-aware alignment. We find that exponential transformation with different $t$'s are good for BoN and WoN procedures, which will be our focus in practical experiments.

### 4.4. Practical implementation of `InfAlign-CTRL`

When implementing `InfAlign-CTRL` in practice, two questions remain to be resolved: (1) How to obtain calibrated reward $\mathcal{C}_{r,\pi_{\mathrm{ref}}}$; (2) How to solve the RLHF problem. To obtain $\mathcal{C}_{r,\pi_{\mathrm{ref}}}$, we consider *empirical calibration*, where we draw $K$ samples $z_1, z_2, ..., z_K$ from the reference model $\pi_{\mathrm{ref}}$ for each prompt $x$ in the RL training data. We then sort the rewards to all the responses $\{r(x, z_1), r(x, z_2), ..., r(x, z_K)\}$, and assign *empirical calibrated reward* scores during RLHF training for the prompt, response pair $(x, y)$ as

$$\widehat{\mathcal{C}}_{r,\pi_{\mathrm{ref}}}(x, y) = \frac{1}{K} \sum_{i=1, z_i \sim \pi_{\mathrm{ref}}}^{K} w_r(y, z_i \mid x). \quad (12)$$

The following lemma establishes the error of approximating $\mathcal{C}_{r,\pi_{\mathrm{ref}}}$ with empirical calibration. The proof follows from DKW inequality (Dvoretzky et al., 1956).

**Lemma 6.** *Given $x \in \mathcal{X}$, for all $\delta > 0$, with probabiltiy at least $1 - \delta$,*

$$\max_{y \in \mathcal{Y}} |\widehat{\mathcal{C}}_{r,\pi_{\mathrm{ref}}}(x, y) - \mathcal{C}_{r,\pi_{\mathrm{ref}}}(x, y)| \leq \sqrt{\frac{\log(2/\delta)}{2K}}.$$

For solving RLHF, we use PPO (Schulman et al., 2017) as the optimization algorithm in this paper to demonstrate the effectiveness of reward transformation. We believe the method could benefit from other advancements of RLHF algorithms.

---

**Algorithm 1** Implementation of `InfAlign-CTRL`

---

**Require:** Base policy $\pi_{\mathrm{ref}}$, (uncalibrated) reward model $r$, set of training prompts $D \subset \mathcal{X}$, number of offline rollouts per prompt $K$, transformation function $\Phi$.
1: Compute empirical calibrated reward $\widehat{\mathcal{C}}_{r,\pi_{\mathrm{ref}}}$ using Eq. (12) with $K$ offline rollouts per $x \in D$.
2: Transform calibrated reward using function $\Phi$ to get $\mathcal{R}_\Phi = \Phi \circ \widehat{\mathcal{C}}_{r,\pi_{\mathrm{ref}}}$
3: Optimize RLHF using calibrated and transformed reward per Eq. (4) using PPO.

---

## 5. Experiment results

### 5.1. Evaluation setup

**Datasets.** We consider the following tasks: (1) Anthropic Helpfulness and Harmlessness datasets (Bai et al., 2022), which involve multi-turn dialogues between a human and a digital assistant. For training the reward models, the preference datasets consist of two responses for one context, and a label for the human preference for the response. We use the train split of the two datasets (44K examples for helpfulness and 42K for harmlessness) to train the uncalibrated and calibrated reward models – separate reward models for each objective. (2) Similarly, for the summarization quality task, we use Reddit posts from TL;DR dataset (Stiennon et al., 2020) and train uncalibrated and calibrated reward models on the train split.

**Model.** The uncalibrated reward model is trained based on the Bradley-Terry pairwise objective (Raffel et al., 2020), and the calibration is done on the training-split of the RL training procedure. The underlying model for both these rewards is the PaLM-2 S model (Anil et al., 2023). The base reference policy model is a PaLM-2 S model that is finetuned (SFT) on the preferred responses of the Anthropic dialog and Reddit summarization datasets. We use `InfAlign-CTRL` with exponential transformations Eq. (11) and PPO as discussed in Algorithm 1 to obtain the aligned policy. We set $K = 100$ in our experiments, and analyze the additional computational overhead in the Appendix.

**Baselines.** We compare against uncalibrated (a model trained to solve RLHF with using the uncalibrated reward model using PPO), BoNBoN (Gui et al., 2024), BoND (Sessa et al., 2024), and IPO (Azar et al., 2023) as baselines.

**True rewards.** As evaluating using ground truth rewards in a pointwise manner based on human annotations can be expensive, we follow (Eisenstein et al., 2024; Mudgal et al., 2024) and perform automated evaluation using a larger PaLM-2 M model to compute *true rewards* and report the win-rate based on these rewards for responses generated by the aligned and base policy models. We acknowledge that there is a gap between these model-generated preference and human preference, but note that prior work (Bai et al., 2022; Stiennon et al., 2020) have reported inter-human-rater agreement on the 3 tasks is often less than 77% and correspondingly the model-generated *true rewards* have accuracy of (77.7%, 77.0, and 76.4%) on the Anthropic Helpfulness, Harmlessness, and Reddit text summarization preference datasets, comparable to state-of-the-art in RewardBench leaderboard (Lambert et al., 2025).

**Metrics.** To measure improvement due to post-RL training, we report both the win rate and the BoN and WoN win rates, along with the corresponding KL-divergence of the RL model with the SFT model. For each of the runs, we experiment with different KL-regularizer strengths ($\beta \in \{0.01, ..., 0.09\}$) and obtain the Pareto-curve of the KL divergence vs {standard, BoN, WoN } win rate curves[5].

---

[5]We use win-rate as our main evaluation metric. We present raw reward comparison for some experiments in Fig. 9 in the appendix.

## 5.2. Reward models are typically miscalibrated

We first show that reward models used on real-world tasks are miscalibrated. We measure the miscalibration of the reward model trained on Anthropic helpfulness preference dataset by computing the scores of 100 reference-policy responses for 10 random prompts from the test split. We then sort the scores and compute the ranks corresponding to each of the responses and plot these values as a scatter plot in Figure 3 (left). If the model were perfectly calibrated, the points for each prompt would lie on the line $y = x$. However, observe that for most prompts, the scatter plot deviates significantly from the $y = x$ line, and the extent of this deviation varies depending on the prompt.

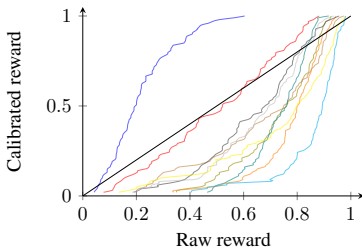

*Figure 3.* Results on reward models trained on the Anthropic helpfulness preference dataset. Scatter plot of reward scores and reward ranks on a random sample of 10 prompts in the Anthropic helpfulness dataset. Note that the model shows miscalibration on most prompts, with the degree of miscalibration varying by prompt.

## 5.3. **InfAlign-CTRL** improves standard win rate

We first measure the performance of InfAlign-CTRL when there is no inference-time procedure applied. In this case, the optimization objective is *standard win rate*, and we compare the performance of InfAlign-CTRL (using an identity transform) against other relevant reward optimization baselines that are known to be (almost) win rate optimal, such as IPO and BoN distillation methods, including BoNBoN, and BoND. In Figure 4, we find InfAlign-CTRL achieves better win rate-KL trade-offs for the Anthropic helpfulness and harmlessness tasks and is on-par with SOTA methods for the Summarization quality task. This demonstrates the advantage of InfAlign-CTRL in the standard win rate setting. We attribute this gain to the properties of the reward calibration step discussed in Section 4. Specifically, calibrated reward is more robust and reward calibration could help mitigate reward hacking (see Appendix D for discussion and results).

## 5.4. **InfAlign-CTRL** improves **BoN**

For the helpfulness objective in the Anthropic dialog dataset, and Reddit summarization quality dataset, we aim to optimize the Best-of-$N$ performance of the aligned model through the exponential transformation of the calibrated rewards. We measure the Best-of-$N$ win rate against the base policy. In Figure 4, we present the result for $N = 4$. We

see that InfAlign-CTRL with exponential transformation achieves up to 3% higher Best-of-$N$ win rates on helpfulness objective, and up to 8% on the summarization quality objective compared to the best SOTA method. As expected, the exponential transformation of the calibrated reward with $t = 10$ outperforms the rest of the models, corroborating the findings on a toy-setting (see Section 4).

## 5.5. **InfAlign-CTRL** improves against **BoN** jailbreaks

For the harmlessness objective in the Anthropic dialog dataset, we aim to improve the Worst-of-$N$ performance of the aligned policy model to improve safety against adversarial actors (Hughes et al., 2024b). Here, we use the negative exponential transformation $t < 0$. In Figure 4 we see that calibration based on the median rewards per-prompt achieves up to 5% higher Worst-of-$N$ win rates as compared to the best SOTA method. The negative transformation of the calibrated reward outperforms the rest of the models, with $t = -10$ performing the best: again identified as the optimal value per our simulation in a toy setting (see Section 4).

## 5.6. Transformation choice for different values of $N$

We further empirically study the choice of transformation on varying values of $N(= 2, 4, 32)$, using BoN as an example (see Figure 5). Within the family of exponential transformations $\Phi(u) = e^{tu}$, we can choose $t$ efficiently using analytical tools with closed form expressions on KL and win rate (Figure 3) without retraining the model. In Figure 5, we see that consistent with our analytical analysis (middle row in Figure 11), $e^{5u}$ achieves better win-rate for Best-of-2, while $e^{10u}$ is better for Best-of-4, demonstrating the transferability of our analytical tool. Moreover, both of these transformations consistently outperforms other non-CTRL baselines for all values of $N$, which demonstrates that they do not overfit to a particular value of $N$.

## 5.7. Calibration helps close the RL optimality gap

To understand how much more performance improvement may still be squeezed out of RL, we compare against BoN (considered as an alignment technique, not inference-time method), which is known to be almost optimal for standard win-rate vs KL divergence tradeoffs (Beirami et al., 2024; Yang et al., 2024; Gui et al., 2024). Concurrent work of GRPO (Shao et al., 2024) shows that calibrating the per-prompt rewards with multiple online rollouts achieves favorable standard win-rate vs KL divergence tradeoffs. In Figure 6, we show that our method InfAlign-CTRL, which relies on offline calibration, performs similar to GRPO in terms of standard win-rate vs KL divergence tradeoffs, without paying the cost of additional online rollouts (more details in Appendix E.1). This demonstrates that the calibration step in our method could be of independent interest in closing

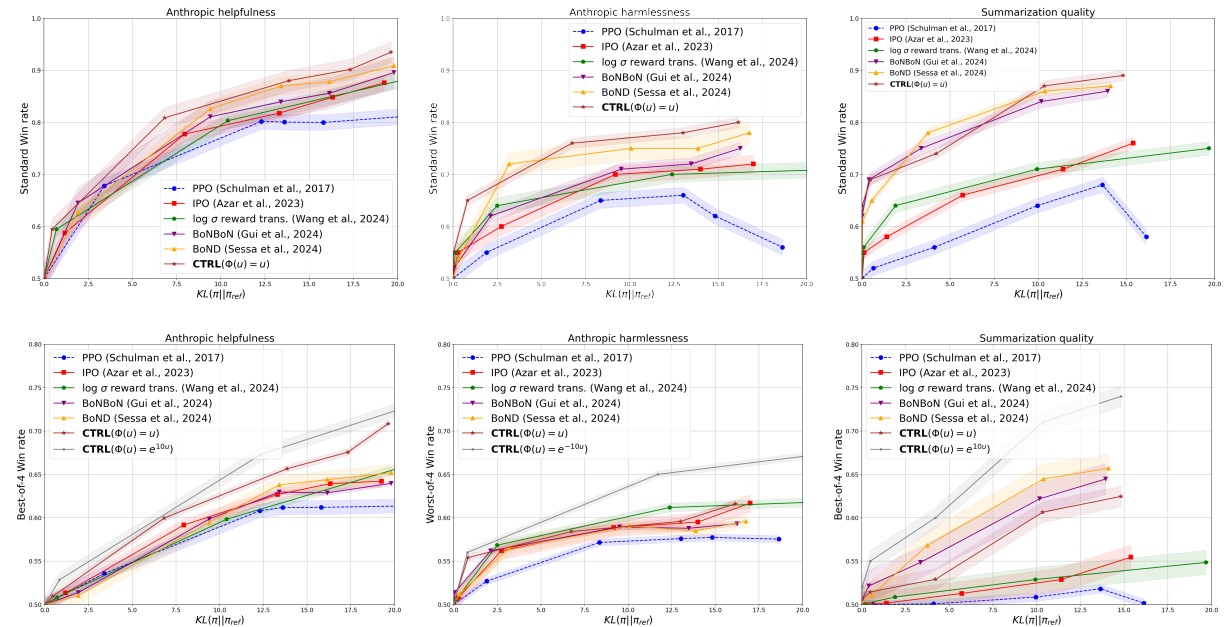

*Figure 4.* (*Top row*) Standard win rate comparison of `InfAlign-CTRL` using identity transformation with other SOTA methods on Anthropic helpfulness, harmlessness, and Reddit summarization dataset. (*Bottom row*) Best/Worst-of-$N$ win rate comparison of `InfAlign-CTRL` using exponential reward transformation. We report win rate against on the test split as measured by the PaLM-2 M reward model trained on the corresponding datasets.

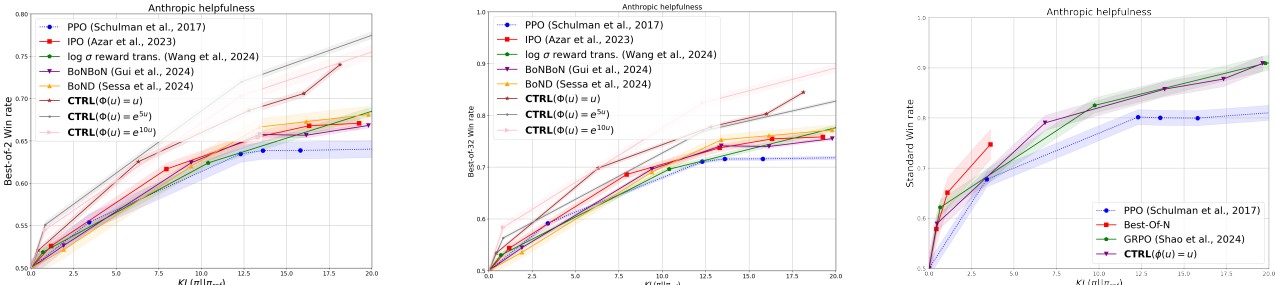

*Figure 5.* Best-of-$N$ win rate comparison on the Anthropic helpfulness dataset with $N = 2, 32$ for different alignment methods.

*Figure 6.* InfAlign-CTRL performs similarly to GRPO closing the gap with BoN.

the optimality gap of RL.

# 6. Concluding Remarks

In this paper, we show that existing win rate optimal RLHF framework suffers from a train-test mismatch when inference-time procedures other than sampling are used. We study the question of how to learn inference-aware optimally aligned language models to close this gap. While learning optimal solutions for general inference-time procedures seems intractable, we propose `InfAlign` — a framework that optimizes for inference-time win rate, and provide theoretical guarantees of finding an optimal inference-aware aligned model. Our framework generalizes prior work on win rate optimal solutions (Azar et al., 2023; Gui et al., 2024) to consider inference-time procedures. We show that for any inference-time procedure, such an optimal model can be learned through the RLHF optimization

framework using reward transformation. We specifically derive transformations for the popular Best-of-$N$ sampling (`BoN`) and jailbreaking (`WoN`) inference-time procedures. We demonstrate the efficacy of this framework, by transferring findings from empirical simulation to real-world tasks and propose `InfAlign-CTRL` — a calibrate-and-transform reinforcement learning solver for ranking based inference-time procedures. Empirically, we demonstrate that in the standard setting when no inference-time procedure is applied, `InfAlign-CTRL` with identity reward transformation achieves slightly better performance compared to a variety of SOTA methods for optimizing standard win rate. When inference-time procedures are applied, we outperform inference-time win rate vs KL tradeoffs compared to existing preference optimization methods by 3-8%.

## Acknowledgment

The authors thank Gholamali Aminian and Suhas Diggavi for their feedback on an earlier version of this work.

## Impact Statement

This paper presents `InfAlign`, a framework whose goal is to advance the field of language model alignment. The framework was designed to boost the performance of alignment in view of different inference-time procedures. While our goal and experiments in this paper were designed to either improve helpfulness or defend against jailbreaks, we acknowledge that bad actors could make use of our framework for aligning models towards adversarial goals. However, this is a possibility that equally applies to any work that advances the field of language model alignment, and we believe that the positive implications of publishing our work outweigh the potential negative implications.

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

# A. Related work

**Inference-time compute.** Test-time compute has been leveraged in recent work (Snell et al., 2024; Brown et al., 2024; Wu et al., 2024a) to achieve better win rate vs KL tradeoffs from the aligned models including controlled decoding (Mudgal et al., 2024; Chakraborty et al., 2024), Monte Carlo tree search (Chaffin et al., 2022; Scialom et al., 2021; Zhao et al., 2024), iterative jailbreak query refinement (Chao et al., 2023), constrained generation (Kim et al., 2025), and model-chaining within agentic frameworks (Gur et al., 2024). Best-of-N (BoN) is also used as an evaluation metric in code and natural language generation benchmarks (Stiennon et al., 2020; Chen et al., 2021). Further, Worst-of-N (WoN) is a popular jailbreaking strategy for adversarial actors to elicit unsafe text from large language models (Hughes et al., 2024b). Prior work has largely focused on approximating inference-time solutions during training time through sampling (Gui et al., 2024; Amini et al., 2024), distillation (Sessa et al., 2024), and decoding (Qiu et al., 2024). Our work is orthogonal to this body of work as they assume that no inference-time procedure is applied, but rather attempt to approximate it during training. We show that our theoretical framework generalizes IPO (Azar et al., 2023) and best-of-$N$ distillation (Gui et al., 2024; Amini et al., 2024; Sessa et al., 2024) as special cases.

We are motivated by recent work that apply meta-generation procedures (Welleck et al., 2024) at inference-time such as chaining prompted models (Brown et al., 2024), problem decomposition through chain-of-thought (Wei et al., 2022), Best-of-N reranking (Collins & Koo, 2005; Charniak & Johnson, 2005; Pauls & Klein, 2009) applied on reasoning traces (OpenAI, 2024). Our `InfAlign` framework was also motivated by complex inference-time strategies that involve transformation techniques such as refinement (Madaan et al., 2024), majority voting (Wang et al., 2022b), or using the generator as input to other search algorithms (Yao et al., 2024), that have outperformed other models for harder tasks. In this spirit, our framework allows to get additional gains in aligning models with such inference-time procedures deployed in the future.

Recent work of Chow et al. (2024) considers inference-aware fine-tuning for BoN sampling. Their study mainly focuses on the binary reward case, which is tailored for math/reasoning tasks. Moreover, there is no KL regularization in the RL objective, making the optimal solution degenerate. This is unsuitable for alignment tasks where it is important to preserve the capability of LLM on other tasks.

**Reward miscalibration.** Reward miscalibration or hacking has been studied extensively in recent work (Amodei et al., 2016; Pang et al., 2022; Gao et al., 2023). The hypotheses behind reward hacking can be broadly categorized into 3 themes: (1) reward underspecification, (2) training-serving skew between pairwise and pointwise reward models, (3) dominant reward due to adhoc transformations. Reward models suffer from *underspecification* due to under-specified training data (Skalse et al., 2022) by capturing spurious correlations in the data (Pan et al., 2022). Methods to mitigate this often include training on non-overlapping splits during reward model fine-tuning (Bai et al., 2022) and ensembling (Coste et al., 2023; Eisenstein et al., 2024). Our `InfAlign-CTRL` method can be easily augmented with such data interventions in reward learning.

**RLHF solvers.** Training reward models on pairwise preference data, and then using it as pointwise scorers during reinforcement learning poses problems of transitive inconsistency. To mitigate this problem, optimization techniques that directly incorporate the pairwise preference data during offline reinforcement learning have been proposed (Rafailov et al., 2023; Azar et al., 2023). Further, calibrating model probabilities to reflect rank-order generated sequences by quality metrics have been proposed (Zhao et al., 2022). We share the motivation behind these methods, while additionally recognizing the need to calibrate the rewards against the base policy on which we are aligning.

When aligning language models for multiple objectives, aggregating the rewards via a weighted sum (Bai et al., 2022; Wu et al., 2024b) is known to result in reward hacking of one of the dominant rewards. Thresholding the effect of individual rewards (Moskovitz et al., 2023) or changing the weights of the training data (Bai et al., 2022), however requires costly hyper-parameter fine-tuning and retraining without the ability to reason about the hyperparameters and their effects on the reward-tradeoffs. Reward transformation techniques that calibrate against a reference reward is effective at mitigating domination of one reward (Wang et al., 2024), but implicitly assumes that the reward aggregation function is a logical "AND" of all rewards, heavily penalizing under-performance on any of the rewards. Motivated by the success of exponential tilting for focusing on high/low quantiles (Li et al., 2023), we also show that `InfAlign-CTRL` with exponential reward transformation achieves near-optimal inference-time win rate vs KL divergence tradeoffs, surpassing the performance of methods such as IPO (Azar et al., 2023) that target to optimize standard win rate vs KL divergence tradeoffs. In this paper, we show that calibration as a first-step can help ground reward transformations based on the final inference-time procedure applied. Further, we build on recent work that show the theoretical guarantees of Best-of-$N$ sampling (Beirami et al., 2024; Gao et al., 2023; Mudgal et al., 2024; Mroueh, 2024) over most reinforcement learning optimization techniques to ground

our calibration and transformation method.

## B. Missing proofs

Many of the results to be proved in this section are under the assumption of continuous models. In this case, $\mathcal{Y}$ can be mapped to $[0, 1]$ through a CDF inverse transformation (Rosenblatt, 1952), with the ordering determined by the reward of the outcomes from the smallest reward to the highest reward. We define the quantile mapping, which is the inverse of the calibrated reward mapping $\mathcal{C}^{-1}$, satisfying $\forall u \in [0, 1]$,

$$\mathcal{C}_{r,\pi,\boldsymbol{x}}^{-1}(u) = \boldsymbol{y}_{u,\boldsymbol{x}} \quad \text{where} \quad \mathcal{C}_{r,\pi}(\boldsymbol{x}, \boldsymbol{y}_{u,\boldsymbol{x}}) = u. \tag{13}$$

Moreover, for these models, we add an additional $\frac{1}{2}\mathbb{1}\{r(\boldsymbol{x}, \boldsymbol{y}) = r(\boldsymbol{x}, \boldsymbol{z})\}$ to the win r.v for simplicity, making $\mathcal{C}_{r,\pi}$ the same as the CDF function.

### B.1. Proof of Lemma 1

We start by stating the policy obtained by KL-regularized RL problem, which can be obtained by standard arguments in the literature (e.g., (Korbak et al., 2022; Rafailov et al., 2023; Yang et al., 2024)).

**Lemma 7.** *The solution to the optimization problem in Definition 1 satisfies*

$$\pi_{r,\beta}^*(\boldsymbol{y} \mid \boldsymbol{x}) \propto \pi_{\text{ref}}(\boldsymbol{y} \mid \boldsymbol{x}) \exp\left(\frac{r(\boldsymbol{x}, \boldsymbol{y})}{\beta}\right).$$

*Proof of Lemma 1.* Let $\pi_{\mathcal{T}}^*$ be a solution to the optimization problem in Eq. (3). Note that for all $\boldsymbol{y}$ such that $\pi_{\mathcal{T}}^*(\boldsymbol{y} \mid \boldsymbol{x}) > 0$, we must have $\pi_{\text{ref}}(\boldsymbol{y} \mid \boldsymbol{x}) > 0$ since otherwise the KL divergence would become infinite.

Then setting

$$\mathcal{R}(\boldsymbol{x}, \boldsymbol{y}) = \beta \log(\pi_{\mathcal{T}}^*(\boldsymbol{y} \mid \boldsymbol{x})/\pi_{\text{ref}}(\boldsymbol{y} \mid \boldsymbol{x}))$$

in Eq. (4) leads to the solution $\pi_{\mathcal{T}}^*$ being its optimal solution as shown in Lemma 7. $\square$

### B.2. Proof of Theorem 1

We first prove Eq. (2), restated below.

$$W_r^{\mathcal{T}}(\pi_1 \succ \pi_2 \mid \boldsymbol{x}) = \sum_{\boldsymbol{y}} \mathcal{C}_{r,\mathcal{T}_{\pi_2}}(\boldsymbol{x}, \boldsymbol{y})\mathcal{T}_{\pi_1}(\boldsymbol{y} \mid \boldsymbol{x}),$$

*Proof.* Proof of Eq. (2) The proof follows from the following identities:

$$\begin{aligned} W_r^{\mathcal{T}}(\pi_1 \succ \pi_2 \mid \boldsymbol{x}) &= \mathbb{E}_{\boldsymbol{z} \sim \mathcal{T}_{\pi_1}(\cdot|\boldsymbol{x}), \boldsymbol{y} \sim \mathcal{T}_{\pi_2}(\cdot|\boldsymbol{x})} \{w_r(\boldsymbol{z}, \boldsymbol{y} \mid \boldsymbol{x})\} \\ &= \mathbb{E}_{\boldsymbol{z} \sim \mathcal{T}_{\pi_1}(\cdot|\boldsymbol{x})} \left\{ \mathbb{E}_{\boldsymbol{y} \sim \mathcal{T}_{\pi_2}(\cdot|\boldsymbol{x})} \{w_r(\boldsymbol{z}, \boldsymbol{y} \mid \boldsymbol{x})\} \right\} \\ &= \mathbb{E}_{\boldsymbol{z} \sim \mathcal{T}_{\pi_1}(\cdot|\boldsymbol{x})} \left\{ \mathcal{C}_{r,\mathcal{T}_{\pi_2}}(\boldsymbol{x}, \boldsymbol{z}) \right\} \\ &= \sum_{\boldsymbol{z}} \mathcal{C}_{r,\mathcal{T}_{\pi_2}}(\boldsymbol{x}, \boldsymbol{z})\mathcal{T}_{\pi_1}(\boldsymbol{z} \mid \boldsymbol{x}). \end{aligned}$$

$\square$

*Proof of Theorem 1.* Note that Eq. (3) has an implicit constraint that $\pi(\cdot \mid \boldsymbol{x})$ must be a valid distribution, i.e.

$$\sum_{\boldsymbol{y}} \pi(\boldsymbol{y} \mid \boldsymbol{x}) = 1.$$

Hence adding the Lagrangian multiplier, we get the following Lagrangian form

$$\mathcal{L}(\pi(\cdot \mid \boldsymbol{x}), \alpha) = W_r^{\mathcal{T}}(\pi \succ \pi_{\text{ref}} \mid \boldsymbol{x}) - \beta D_{\text{KL}}(\pi(\cdot \mid \boldsymbol{x})\|\pi_{\text{ref}}(\cdot|\boldsymbol{x})) + \alpha\left(\sum_{\boldsymbol{y}} \pi(\boldsymbol{y} \mid \boldsymbol{x}) - 1\right).$$

By method of Lagrange multipliers, we have that the solution to Eq. (3) must be a stationary point of $\mathcal{L}(\pi(\cdot \mid \boldsymbol{x}), \alpha)$. And hence

$$\boldsymbol{0} = \frac{\partial \mathcal{L}(\pi(\cdot \mid \boldsymbol{x}), \alpha)}{\partial \pi(\boldsymbol{y} \mid \boldsymbol{x})} = \frac{\partial W_r^{\mathcal{T}}(\pi \succ \pi_{\mathrm{ref}} \mid \boldsymbol{x})}{\partial \pi(\boldsymbol{y} \mid \boldsymbol{x})} - \beta \left( \log \frac{\pi(\boldsymbol{y} \mid \boldsymbol{x})}{\pi_{\mathrm{ref}}(\boldsymbol{y} \mid \boldsymbol{x})} + 1 \right) + \alpha.$$

Setting

$$\mathcal{R}(\boldsymbol{x}, \boldsymbol{y}) = \frac{\partial W_r^{\mathcal{T}}(\pi \succ \pi_{\mathrm{ref}} \mid \boldsymbol{x})}{\partial \pi(\boldsymbol{y} \mid \boldsymbol{x})},$$

we get

$$\log \frac{\pi(\boldsymbol{y} \mid \boldsymbol{x})}{\pi_{\mathrm{ref}}(\boldsymbol{y} \mid \boldsymbol{x})} = \frac{\mathcal{R}(\boldsymbol{x}, \boldsymbol{y}) + \alpha}{\beta} - 1.$$

And hence

$$\pi(\boldsymbol{y} \mid \boldsymbol{x}) \propto \pi_{\mathrm{ref}}(\boldsymbol{y} \mid \boldsymbol{x}) \exp \left( \frac{\mathcal{R}(\boldsymbol{x}, \boldsymbol{y})}{\beta} \right).$$

It remains to prove Eq. (7). Plugging in $\pi_1 = \pi$, $\pi_2 = \pi_{\mathrm{ref}}$ to Eq. (2), and taking partial derivative with respect to $\pi(\boldsymbol{y} \mid \boldsymbol{x})$ on the right hand side completes the proof. $\qquad\square$

### B.3. Proof of Lemma 2

If $r(\boldsymbol{x}, \boldsymbol{y}) \geq r(\boldsymbol{x}, \boldsymbol{z})$, we have $\forall \boldsymbol{y}'$,

$$w_r(\boldsymbol{y}, \boldsymbol{y}' \mid \boldsymbol{x}) \geq w_r(\boldsymbol{z}, \boldsymbol{y}' \mid \boldsymbol{x}).$$

Hence

$$\mathcal{C}_{r,\pi_{\mathrm{ref}}}(\boldsymbol{x}, \boldsymbol{y}) = \mathbb{E}_{\boldsymbol{y}' \sim \pi(\cdot \mid \boldsymbol{x})} w_r(\boldsymbol{y}, \boldsymbol{y}' \mid \boldsymbol{x}) \geq \mathbb{E}_{\boldsymbol{y}' \sim \pi(\cdot \mid \boldsymbol{x})} w_r(\boldsymbol{z}, \boldsymbol{y}' \mid \boldsymbol{x}) = \mathcal{C}_{r,\pi_{\mathrm{ref}}}(\boldsymbol{x}, \boldsymbol{z}).$$

### B.4. Proof of Lemma 3

The proof follows from the fact that for all $\boldsymbol{x}$ and $\boldsymbol{y}$

$$
\begin{aligned}
\mathcal{C}_{g(r),\pi_{\mathrm{ref}}}(\boldsymbol{x}, \boldsymbol{y}) &= \mathbb{E}_{\boldsymbol{z} \sim \pi_{\mathrm{ref}}} \left\{ \mathbf{1}[g(r(\boldsymbol{x}, \boldsymbol{y})) > g(r(\boldsymbol{x}, \boldsymbol{z}))] + \frac{1}{2} \mathbf{1}[g(r(\boldsymbol{x}, \boldsymbol{y})) = g(r(\boldsymbol{x}, \boldsymbol{z}))] \right\} \\
&= \mathbb{E}_{\boldsymbol{z} \sim \pi_{\mathrm{ref}}} \left\{ \mathbf{1}[r(\boldsymbol{x}, \boldsymbol{y}) > r(\boldsymbol{x}, \boldsymbol{z})] + \frac{1}{2} \mathbf{1}[r(\boldsymbol{x}, \boldsymbol{y}) = r(\boldsymbol{x}, \boldsymbol{z})] \right\} \qquad (14) \\
&= \mathcal{C}_{r,\pi_{\mathrm{ref}}}(\boldsymbol{x}, \boldsymbol{y}),
\end{aligned}
$$

where (14) follows from monotone increasing property of $g$.

### B.5. Proof of Lemma 4

To show that $\mathcal{C}_{r,\pi_{\mathrm{ref}}}(\boldsymbol{x}, \boldsymbol{y}) \sim \mathrm{Unif}([0, 1])$, it would be enough to show $\forall u \in [0, 1]$,

$$\Pr_{\boldsymbol{y} \sim \pi_{\mathrm{ref}}} \left( \mathcal{C}_{r,\pi_{\mathrm{ref}}}(\boldsymbol{x}, \boldsymbol{y}) \leq u \right) = u.$$

Recall the definition of $\boldsymbol{y}_{u,\boldsymbol{x}}$ in Eq. (13), we have that

$$
\begin{aligned}
\Pr_{\boldsymbol{y} \sim \pi_{\mathrm{ref}}} \left( \mathcal{C}_{r,\pi_{\mathrm{ref}}}(\boldsymbol{x}, \boldsymbol{y}) \leq u \right) &= \Pr_{\boldsymbol{y} \sim \pi_{\mathrm{ref}}} \left( r(\boldsymbol{x}, \boldsymbol{y}) \leq r(\boldsymbol{x}, \boldsymbol{y}_{u,\boldsymbol{x}}) \right) \\
&= \mathcal{C}_{r,\pi_{\mathrm{ref}}}(\boldsymbol{x}, \boldsymbol{y}_{u,\boldsymbol{x}}) \\
&= u,
\end{aligned}
$$

completing the proof.

## B.6. Proof of Lemma 5

For continuous language models, we have $\text{BoN}_\pi$ satisfies that $\forall \boldsymbol{x}, \boldsymbol{y}$

$$\text{Pr}_{\boldsymbol{z} \sim \text{BoN}_\pi(\cdot | \boldsymbol{x})} \left( r(\boldsymbol{x}, \boldsymbol{y}) \leq r(\boldsymbol{x}, \boldsymbol{y}) \right) = \text{Pr}_{\boldsymbol{z} \sim \pi(\cdot | \boldsymbol{x})} \left( r(\boldsymbol{x}, \boldsymbol{z}) \leq r(\boldsymbol{x}, \boldsymbol{y}) \right)^N = \mathcal{C}_{r, \pi_{\text{ref}}}(\boldsymbol{x}, \boldsymbol{y})^N. \tag{15}$$

Hence

$$\text{BoN}_\pi(\boldsymbol{y} \mid \boldsymbol{x}) = \frac{\text{dPr}_{\boldsymbol{z} \sim \text{BoN}_\pi(\cdot | \boldsymbol{x})} \left( r(\boldsymbol{x}, \boldsymbol{z}) \leq r(\boldsymbol{x}, \boldsymbol{y}) \right)}{\text{d}\boldsymbol{y}} = \pi(\boldsymbol{y} \mid \boldsymbol{x}) \mathcal{C}_{r, \pi_{\text{ref}}}(\boldsymbol{x}, \boldsymbol{y})^{N-1}.$$

For $\text{WoN}_\pi$, we have

$$\text{Pr}_{\boldsymbol{z} \sim \text{WoN}_\pi(\cdot | \boldsymbol{x})} \left( r(\boldsymbol{x}, \boldsymbol{y}) \leq r(\boldsymbol{x}, \boldsymbol{y}) \right) = 1 - \text{Pr}_{\boldsymbol{z} \sim \pi(\cdot | \boldsymbol{x})} \left( r(\boldsymbol{x}, \boldsymbol{z}) > r(\boldsymbol{x}, \boldsymbol{y}) \right)^N = 1 - \left( 1 - \mathcal{C}_{r, \pi_{\text{ref}}}(\boldsymbol{x}, \boldsymbol{y}) \right)^N. \tag{16}$$

Hence

$$\text{WoN}_\pi(\boldsymbol{y} \mid \boldsymbol{x}) = \frac{\text{dPr}_{\boldsymbol{z} \sim \text{WoN}_\pi(\cdot | \boldsymbol{x})} \left( r(\boldsymbol{x}, \boldsymbol{z}) \leq r(\boldsymbol{x}, \boldsymbol{y}) \right)}{\text{d}\boldsymbol{y}} = \pi(\boldsymbol{y} \mid \boldsymbol{x}) \left( 1 - \mathcal{C}_{r, \pi_{\text{ref}}}(\boldsymbol{x}, \boldsymbol{y}) \right)^{N-1}.$$

## B.7. Proof of Theorem 2

In this section, we will prove an extended version of Theorem 2 below.

**Theorem 4** (Extended version of Theorem 2). *If $\mathcal{T}$ is a calibrated inference-time procedure with mapping function $g_\mathcal{T}$, for any continuous language model $\pi$, $\beta > 0$ and reward transformation function $\Phi$, we have that*

$$W_r^\mathcal{T}(\pi_{\mathcal{R}_\Phi, \beta}^* \succ \pi_{\text{ref}} \mid \boldsymbol{x}) = \frac{\int_0^1 \exp\left(\Phi(u)/\beta\right) g_\mathcal{T}(F_{\Phi, \beta}(u)) \int_0^u g_\mathcal{T}(u') \text{d}u' \text{d}u}{\int_0^1 \exp\left(\Phi(u)/\beta\right) g_\mathcal{T}(F_{\Phi, \beta}(u)) \text{d}u \int_0^1 g_\mathcal{T}(u) \text{d}u}$$

*where $F_{\Phi, \beta}(u) = \frac{\int_0^u e^{\Phi(u')/\beta} \text{d}u'}{\int_0^1 e^{\Phi(u')/\beta} \text{d}u'}$, and*

$$D_{KL}(\pi_{\mathcal{R}_\Phi, \beta}^* \| \pi_{\text{ref}}) = \frac{1}{\beta} \frac{\int_0^1 \Phi(u) e^{\Phi(u)/\beta} \text{d}u}{\int_0^1 e^{\Phi(u)/\beta} \text{d}u} - \log\left(\int_0^1 e^{\Phi(u)/\beta} \text{d}u\right).$$

*And hence they are independent of $r$ and $\pi_{\text{ref}}$.*

*Proof.* In the proof, we consider the two language models $\pi_{\mathcal{R}_\Phi, \beta}^*$ and $\pi_{\text{ref}}$ in the space of calibrated reward against the base policy $\pi_{\text{ref}}$. For any policy $\pi$, let $\mathcal{C}_{r, \pi_{\text{ref}}} \circ \pi(\cdot \mid \boldsymbol{x})$ be a distribution over $[0, 1]$ that outputs the calibrated reward $\mathcal{C}_{r, \pi_{\text{ref}}}(\boldsymbol{x}, \boldsymbol{y})$ of the sample $\boldsymbol{y}$ sampled from $\pi(\cdot \mid \boldsymbol{x})$. Then by Lemma 4,

$$\mathcal{C}_{r, \pi_{\text{ref}}} \circ \pi(\cdot \mid \boldsymbol{x}) \sim \text{Unif}([0, 1]).$$

Similarly, using Lemma 7, it can be shown that $\mathcal{C}_{r, \pi_{\text{ref}}} \circ \pi_{\mathcal{R}_\Phi, \beta}^*$ follows the distribution with density $\forall u \in [0, 1]$,

$$\mathcal{C}_{r, \pi_{\text{ref}}} \circ \pi_{\mathcal{R}_\Phi, \beta}^*(u \mid \boldsymbol{x}) = \frac{e^{\Phi(u)/\beta}}{\int_0^1 e^{\Phi(u')/\beta} \text{d}u'}.$$

Note that since $r$ assigns distinct rewards to different $\boldsymbol{y}$'s and $\mathcal{C}_{r, \pi_{\text{ref}}}$ is a monotone transformation of $r$, we have that

$$
\begin{aligned}
D_{\text{KL}}(\pi_{\mathcal{R}_\Phi, \beta}^* \| \pi_{\text{ref}}) &= D_{\text{KL}}(\mathcal{C}_{r, \pi_{\text{ref}}} \circ \pi_{\mathcal{R}_\Phi, \beta}^* \| \mathcal{C}_{r, \pi_{\text{ref}}} \circ \pi) \\
&= \int_{u=0}^1 \frac{e^{\Phi(u)/\beta}}{\int_0^1 e^{\Phi(u')/\beta} \text{d}u'} \log \frac{e^{\Phi(u)/\beta}}{\int_0^1 e^{\Phi(u')/\beta} \text{d}u'} \text{d}u \\
&= \frac{1}{\beta} \frac{\int_{u=0}^1 \Phi(u) e^{\Phi(u)/\beta} \text{d}u}{\int_0^1 e^{\Phi(u)/\beta} \text{d}u} - \log\left(\int_0^1 e^{\Phi(u)/\beta} \text{d}u\right).
\end{aligned}
$$

After the inference-time procedure $\mathcal{T}$ is applied, we have that inference-time base policy satisfies

$$\mathcal{C}_{r,\pi_{\text{ref}}} \circ \mathcal{T}_{\pi_{\text{ref}}}(u \mid \boldsymbol{x}) = \frac{g_{\mathcal{T}}(u)}{\int_0^1 g_{\mathcal{T}}(u') \mathrm{d}u'}.$$

For the inference-time aligned policy, we have that:

$$\Pr_{\boldsymbol{y} \sim \pi^*_{\mathcal{R}_\Phi,\beta}(\cdot|\boldsymbol{x})} \left( \mathcal{C}_{r,\pi_{\text{ref}}}(\boldsymbol{x},\boldsymbol{y}) \leq u \right) = \frac{\int_0^u e^{\Phi(u')/\beta} \mathrm{d}u'}{\int_0^1 e^{\Phi(u')/\beta} \mathrm{d}u'},$$

which is defined as $F_{\Phi,\beta}(u)$. And hence we have

$$\mathcal{C}_{r,\pi_{\text{ref}}} \circ \mathcal{T}_{\pi^*_{\mathcal{R}_\Phi,\beta}}(u \mid \boldsymbol{x}) = \frac{\exp\left(\Phi(u)/\beta\right) g_{\mathcal{T}}(F_{\Phi,\beta}(u))}{\int_0^1 \exp\left(\Phi(u)/\beta\right) g_{\mathcal{T}}(F_{\Phi,\beta}(u)) \mathrm{d}u}.$$

Thus, the inference-time win rate satisfies

$$
\begin{aligned}
W_r^{\mathcal{T}}(\pi^*_{\mathcal{R}_\Phi,\beta} \succ \pi_{\text{ref}} \mid \boldsymbol{x}) &= \mathbb{E}_{\boldsymbol{y} \sim \mathcal{T}_{\pi^*_{\mathcal{R}_\Phi,\beta}}(\cdot|\boldsymbol{x}), \boldsymbol{z} \sim \mathcal{T}_{\pi_{\text{ref}}}(\cdot|\boldsymbol{x})} \left\{ \mathbb{1}\{r(\boldsymbol{x},\boldsymbol{y}) \geq r(\boldsymbol{x},\boldsymbol{z})\} \right\} \\
&= \mathbb{E}_{\boldsymbol{y} \sim \mathcal{T}_{\pi^*_{\mathcal{R}_\Phi,\beta}}(\cdot|\boldsymbol{x}), \boldsymbol{z} \sim \mathcal{T}_{\pi_{\text{ref}}}(\cdot|\boldsymbol{x})} \left\{ \mathbb{1}\{\mathcal{C}_{r,\pi_{\text{ref}}}(\boldsymbol{x},\boldsymbol{y}) \geq \mathcal{C}_{r,\pi_{\text{ref}}}(\boldsymbol{x},\boldsymbol{z})\} \right\} \\
&= \Pr_{u \sim \mathcal{C}_{r,\pi_{\text{ref}}} \circ \mathcal{T}_{\pi^*_{\mathcal{R}_\Phi,\beta}}(\cdot|\boldsymbol{x}), u' \sim \mathcal{C}_{r,\pi_{\text{ref}}} \circ \mathcal{T}_{\pi_{\text{ref}}}(\cdot|\boldsymbol{x})} (u' \leq u) \\
&= \frac{\int_0^1 \exp\left(\Phi(u)/\beta\right) g_{\mathcal{T}}(F_{\Phi,\beta}(u)) \int_0^u g_{\mathcal{T}}(u') \mathrm{d}u' \mathrm{d}u}{\int_0^1 \exp\left(\Phi(u)/\beta\right) g_{\mathcal{T}}(F_{\Phi,\beta}(u)) \mathrm{d}u \int_0^1 g_{\mathcal{T}}(u) \mathrm{d}u},
\end{aligned}
$$

where the second equality follows from Lemma 2, completing the proof. $\qquad\square$

### B.8. Proof of Theorem 3

The KL divergence is the same as the KL divergence in Theorem 4. The win rate can be obtained by plugging $g_{\text{BoN}}(u) = u^{N-1}$ and $g_{\text{WoN}}(u) = (1-u)^{N-1}$ (as shown in Lemma 5) into Theorem 4.

### B.9. Proof of Corollary 2

We will show that Corollary 2 is a special case of Theorem 1 with a simple continuous language model. And by Theorem 2, we have the $\Phi$ can be generalized to arbitrary continuous language models.

Let $\mathcal{Y} = [0,1]$. We assume the LMs and reward models are context-independent. We use $u \in [0,1]$ to denote $\boldsymbol{y}$ and set the reward model to be $r(u) = u$. The base policy is a simple uniform distribution over $[0,1]$, $\pi_{\text{ref}} = \text{Unif}([0,1])$. Note that $F_\pi(u)$ be the CDF of $\pi$, then we have that the BoN win rate is

$$W_r^{\text{BoN}}(\pi \succ \pi_{\text{ref}} \mid \boldsymbol{x}) = 1 - N \int_0^1 F_\pi(u)^N u^{N-1} \mathrm{d}u,$$

and WoN win rate is

$$W_r^{\text{WoN}}(\pi \succ \pi_{\text{ref}} \mid \boldsymbol{x}) = N \int_0^1 \left(1 - F_\pi(u)\right)^N (1-u)^{N-1} \mathrm{d}u.$$

Plugging these into Theorem 1, we have for BoN,

$$
\begin{aligned}
\mathcal{R}(u) &= \frac{\partial W_r^{\text{BoN}}(\pi \succ \pi_{\text{ref}} \mid \boldsymbol{x})}{\partial \pi(u)} \\
&= -N \int_0^1 v^{N-1} \frac{\partial F_\pi(v)^N}{\partial \pi(u)} \mathrm{d}v \\
&= -N^2 \int_0^1 v^{N-1} F_\pi(v)^{N-1} \frac{\partial F_\pi(v)}{\partial \pi(u)} \mathrm{d}v \\
&= -N^2 \int_0^1 F_\pi(v)^{N-1} \mathbb{1}\{v \geq u\} v^{N-1} \mathrm{d}v \\
&= -N^2 \int_u^1 F_\pi(v)^{N-1} v^{N-1} \mathrm{d}v.
\end{aligned}
$$

For WoN, we have

$$
\begin{aligned}
\mathcal{R}(u) &= \frac{\partial W_r^{\text{WoN}}(\pi \succ \pi_{\text{ref}} \mid \boldsymbol{x})}{\partial \pi(u)} \\
&= N \int_0^1 (1-v)^{N-1} \frac{\partial \left(1 - F_\pi(v)\right)^N}{\partial \pi(u)} \mathrm{d}v \\
&= -N^2 \int_0^1 (1-v)^{N-1} (1 - F_\pi(v))^{N-1} \frac{\partial F_\pi(v)}{\partial \pi(u)} \mathrm{d}v \\
&= -N^2 \int_0^1 (1-v)^{N-1} (1 - F_\pi(v))^{N-1} \mathbb{1}\{v \geq u\} \mathrm{d}v \\
&= -N^2 \int_u^1 (1-v)^{N-1} (1 - F_\pi(v))^{N-1} \mathrm{d}v.
\end{aligned}
$$

## C. The role of KL divergence in model alignment

One question that arises is the role of the KL divergence regularizer in Eq. (3). In this section, we argue that the regularizer essentially enables multi-tasking between the SFT task and the RL task.

Let's consider a log-linear model such that

$$\pi_\theta(\boldsymbol{y}|\boldsymbol{x}) = e^{\theta^T g(\boldsymbol{x},\boldsymbol{y}) - A(\theta;\boldsymbol{x})}, \tag{17}$$

where $g(\boldsymbol{x},\boldsymbol{y})$ is a fixed encoding of $(\boldsymbol{x},\boldsymbol{y})$, and $A(\theta;\boldsymbol{x})$ is the partition function normalizing the distribution.

**Supervised finetuning (SFT).**   Let $D_{\text{sft}}(\boldsymbol{x},\boldsymbol{y}) = \mu(\boldsymbol{x}) \times p_{\text{sft}}(y|\boldsymbol{x})$ be the SFT data distribution. Then, the SFT task is

$$\theta_{\text{sft}}^* = \arg\min_\theta \mathcal{L}_{\text{sft}}(\theta) \qquad \text{where} \qquad \mathcal{L}_{\text{sft}}(\theta) := E_{(\boldsymbol{x},\boldsymbol{y})\sim D_{\text{sft}}}\{A(\theta;\boldsymbol{x}) - \theta^\top g(\boldsymbol{x},\boldsymbol{y})\}, \tag{18}$$

We further call $p = \pi_{\theta_{\text{sft}}^*}$.

**Lemma 8.** *The SFT solution satisfies*

$$E_{x\sim\mu}\{\nabla_\theta A(\theta_{sft}^*)\} = E_{(\boldsymbol{x},\boldsymbol{y})\sim D_{sft}} g(\boldsymbol{x},\boldsymbol{y}). \tag{19}$$

*Proof.* This is a known property of exponential families. The proof follows by noticing $\nabla_\theta \mathcal{L}_{\text{sft}}(\theta_{\text{sft}}^*) = 0$.   □

**KL-regularized reward optimization (RO).**   Let $r$ be a reward function that determines the reward for each $(\boldsymbol{x},\boldsymbol{y})$. Let $\mathcal{L}_{\text{ro}}(\theta) := -E_{\boldsymbol{x}\sim\mu} E_{\boldsymbol{y}\sim\pi_\theta} r(\boldsymbol{x},\boldsymbol{y})$. Then,

$$\theta_{\text{bilevel},\beta}^* = \arg\min_\theta \mathcal{L}_{\text{bilevel},\beta}(\theta) \qquad \text{where} \qquad \mathcal{L}_{\text{bilevel}}(\theta) := D_{\text{KL}}(\pi_\theta \| p) + \frac{1}{\beta}\mathcal{L}_{\text{ro}}(\theta), \tag{20}$$

where $D_{\text{KL}}(\pi_\theta \| p) = E_{\boldsymbol{x}\sim\mu} D_{\text{KL}}(\pi_\theta(\cdot|\boldsymbol{x}) \| p(\cdot|\boldsymbol{x}))$.

**Multi-tasking SFT and RO.**   Now consider the following tasks

$$\theta_{\text{multi-task},\beta}^* = \arg\min_\theta \mathcal{L}_{\text{multi-task},\beta}(\theta) \qquad \text{where} \qquad \mathcal{L}_{\text{multi-task}}(\theta) := \mathcal{L}_{\text{sft}}(\theta) + \frac{1}{\beta}\mathcal{L}_{\text{ro}}(\theta). \tag{21}$$

**Theorem 5.** *For all $\beta \in \mathbb{R}$, we have $\theta_{bilevel,\beta}^* = \theta_{multi\text{-}task,\beta}^*$.*

*Proof.* Notice that

$$\mathcal{L}_{\text{bilevel},\beta}(\theta) = D_{\text{KL}}(\pi_\theta \| p) + \frac{1}{\beta}\mathcal{L}_{\text{ro}}(\theta) \tag{22}$$

$$= E_{x\sim\mu}\{A(\theta;\boldsymbol{x}) - A(\theta_{\text{sft}}^*;\boldsymbol{x}) - (\theta - \theta_{\text{sft}}^*)^\top \nabla_\theta A(\theta_{\text{sft}}^*;\boldsymbol{x})\} + \frac{1}{\beta}\mathcal{L}_{\text{ro}}(\theta) \tag{23}$$

$$= E_{x\sim\mu}\{A(\theta;\boldsymbol{x}) - A(\theta_{\text{sft}}^*;\boldsymbol{x})\} - (\theta - \theta_{\text{sft}}^*)^\top E_{(\boldsymbol{x},\boldsymbol{y})\sim D_{\text{sft}}} g(\boldsymbol{x},\boldsymbol{y}) + \frac{1}{\beta}\mathcal{L}_{\text{ro}}(\theta) \tag{24}$$

$$= \mathcal{L}_{\text{multi-task},\beta}(\theta) + \mathcal{L}_{\text{sft}}(\theta_{\text{sft}}^*), \tag{25}$$

where Eq. (23) follows by noticing that KL divergence is a Bregman divergence in this setup, Eq. (24) follows from Lemma 8, and Eq. (25) follows from the definition of $\mathcal{L}_{\text{sft}}(\theta)$ applied to $\theta$ and $\theta_{\text{sft}}^*$. Hence, the minimizers of the two objectives are the same given that

$$\mathcal{L}_{\text{bilevel},\beta}(\theta) = \mathcal{L}_{\text{multi-task},\beta}(\theta) + C,$$

completing the proof.   □

Thus, effectively this proves that the RLHF objective enables multi-tasking between the SFT stage and the reward optimization RL objective.

One may wonder why we did not pose the KL divergence regularizer on the transformed distributions through $D_{\mathrm{KL}}(\mathcal{T}_\pi(\cdot \mid \boldsymbol{x}) \| \mathcal{T}_{\pi_{\mathrm{ref}}}(\cdot | \boldsymbol{x}))$ instead. Consider the Best-of-$N$ jailbreaking for example. While the adversary may be using the model to generate $N$ responses and choose the least safe one for jailbreaking, the model should possess the core capabilities for other types of inference-time usage for other tasks that is different from that of jailbreaking (e.g., through chain-of-thought). Therefore, changing the KL divergence regularizer does not capture the fact that the model should remain suitable for all other tasks, and not just for the one for which it is getting aligned. We also note that if we used $D_{\mathrm{KL}}(\mathcal{T}_\pi)(\cdot \mid \boldsymbol{x}) \| \mathcal{T}_{\pi_{\mathrm{ref}}}(\cdot | \boldsymbol{x}))$ instead, the problem would actually simplify to the standard alignment problem through a simple change of variables.

# D. Calibration Reduces Reward Hacking

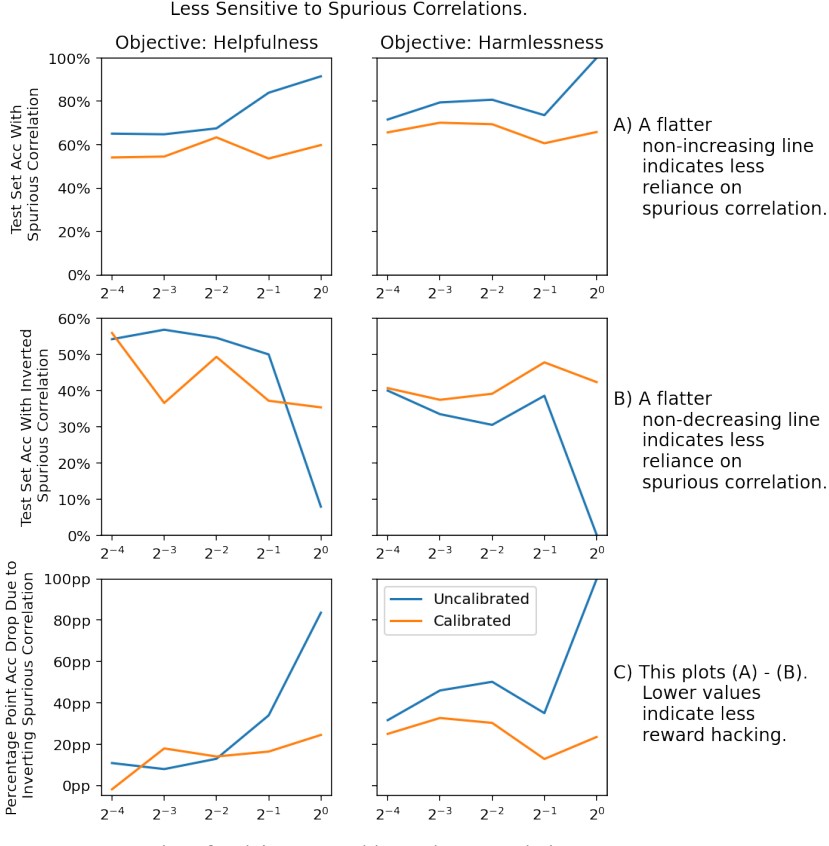

*Figure 7.* Calibrated reward models demonstrate robustness against reward hacking: We poisoned the training data by adding phrases to the preferred response to induce spurious correlations. When we evaluated against a test set where the correlations are inverted (phrase added to unpreferred models), calibrated models maintained higher accuracy than uncalibrated ones, demonstrating their reduced reliance on spurious correlations.

We demonstrate that calibrated reward models are less susceptible to reward hacking, a phenomenon where models exploit spurious correlations in training data to optimize for reward signals instead of true task objectives.

To induce reward hacking, we injected specific phrases into the start of preferred responses of our preference datasets: "Sorry, I can't help you with that" for Harmlessness and "Sure" for Helpfulness. We then evaluated the model's accuracy on a poisoned evaluation set where these phrases were inverted (added to the *unpreferred* responses). A significant drop in accuracy on this poisoned set would indicate reward hacking: a reliance on the spurious correlation.

Figure 7 shows that calibrated reward models are more robust to these manipulated correlations, maintaining higher accuracy compared to uncalibrated models.

# E. Additional experimental results

In Fig. 8, we present the `BoN` and `WoN` win rate comparisons with $N = 32$. We see `InfAlign-CTRL` leads to improvement on the win rate compared to other SOTA methods. For `BoN`, we observe better gains as compared to $N = 4$ on Anthropic helpfulness, and Reddit summarization datasets. This shows the importance of `InfAlign` when more inference-time compute is preformed.

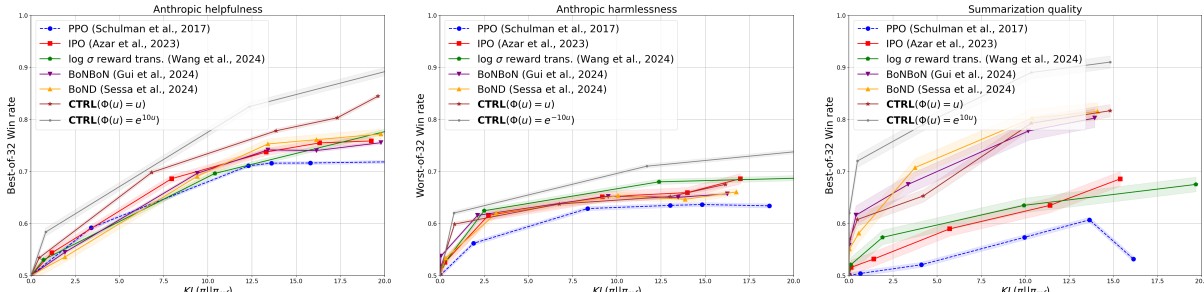

*Figure 8.* Best/Worst-of-32 win rate comparison of `InfAlign-CTRL` using exponential reward transformation. We report win rate against on the test split as measured by the PaLM-2 M reward model trained on the corresponding datasets.

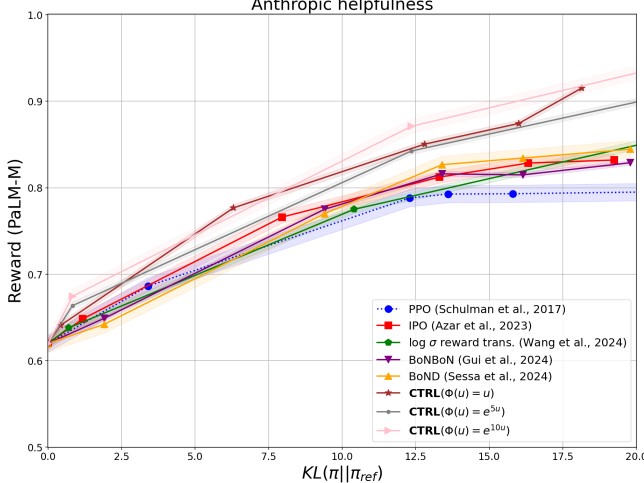

*Figure 9.* Expected raw rewards vs $D_{\mathrm{KL}}(\pi\|\pi_{\mathrm{ref}})$ for the aligned model used to compute the standard win-rate of Anthropic helpfulness dialog task (Fig. 4-top left).

In Fig. 9, we plot the expected raw reward from the judge model (PaLM-M) vs $D_{\mathrm{KL}}(\pi\|\pi_{\mathrm{ref}})$ tradeoff for the aligned model used to compute the standard win-rate of Anthropic helpfulness dialog task (Fig. 4-top left). We see that our proposed algorithm `InfAlign-CTRL` outperforms other baselines in expected raw rewards as well.

## E.1. Comparison with **GRPO**

Group Relative Policy Optimization (Shao et al., 2024) proposes an iterative training setup rolling out multiple samples for a given prompt (group) from the policy model and centering the on-policy rewards to achieve better inference-time reasoning capabilities. We compare against a controlled setting where the base policy of the `GRPO` rollouts is fixed i.e. one iteration of the outer loop of `GRPO`. Further, we train it for a smaller number of epochs (2) with the same number of samples per prompt (K=G=100) to obtain an equivalent total number of reward rollouts during training, and demonstrate that `InfAlign − CTRL` outperforms this version of `GRPO` in the Best-of-4 inference-time evaluation, and comparable to CTRL and BoND in the standard win-rate evaluation (Figure 10). This further demonstrates that CTRL provides an efficient offline reward calibration and transformation alternative to the more compute intensive online reward calibration approaches which are not inference-aware.

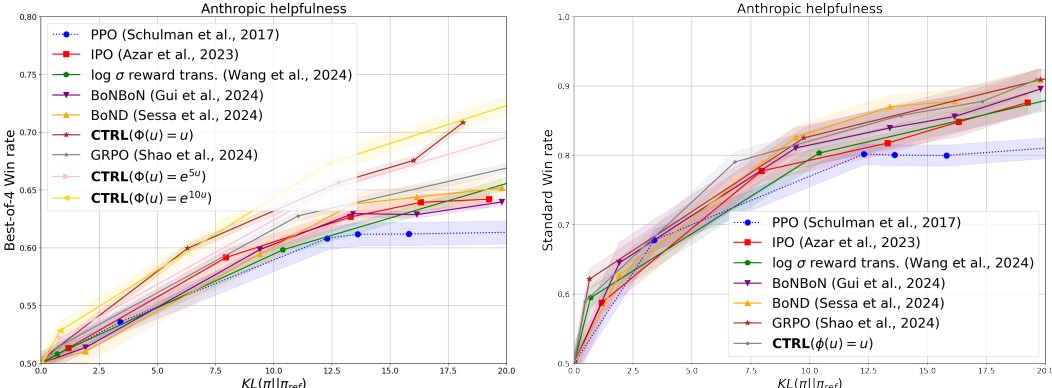

*Figure 10.* CTRL outperforms GRPO in Best-of-4 and comparable in standard win-rate evaluation settings.

## F. Analytical comparison of different transformations

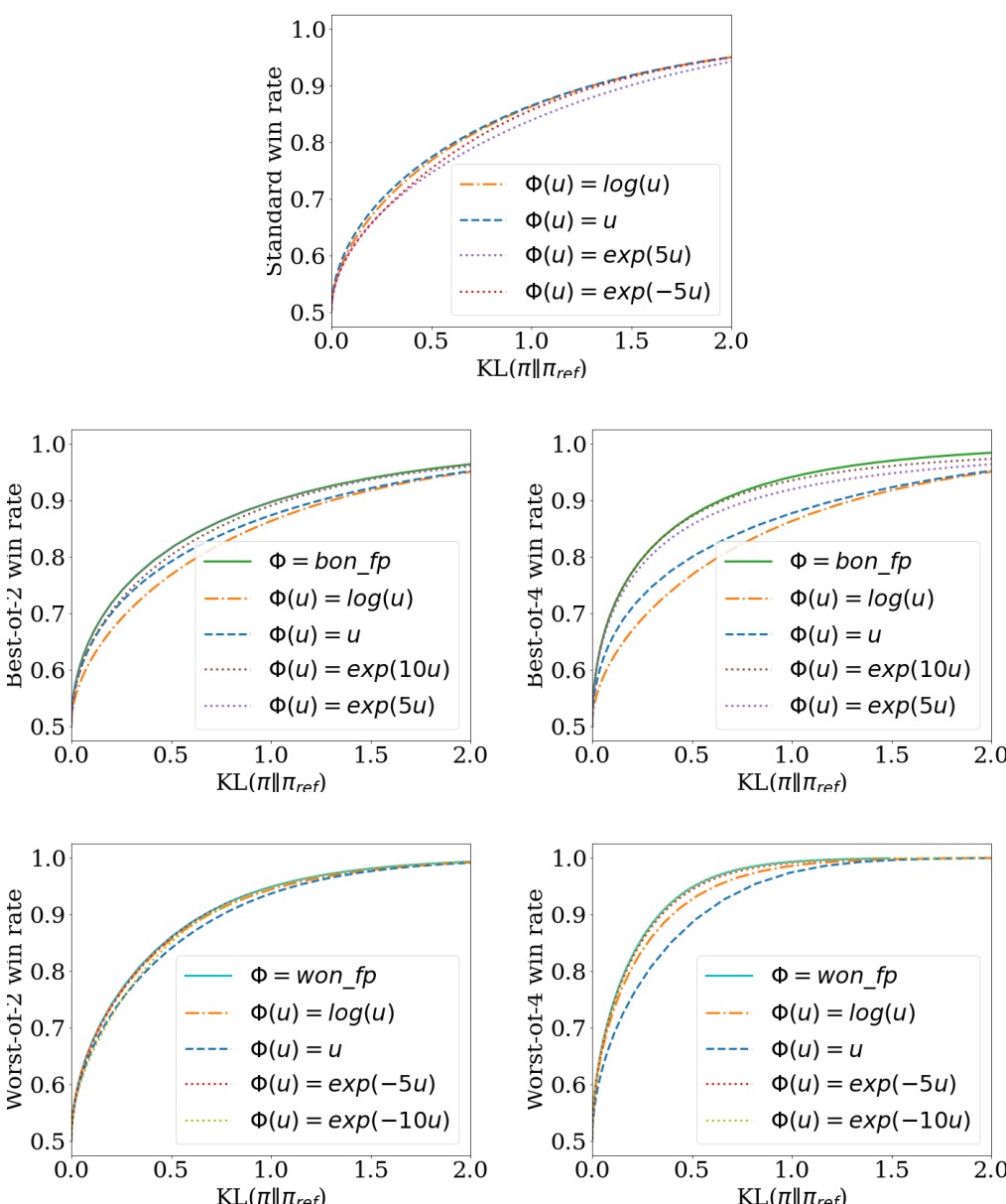

*Figure 11.* Standard, Best-of-$N$, and worst-of-$N$ win rate vs KL tradeoff curves for $N = 2, 4$ with different transformation functions.

In this section, we give an extended discussion of analytical results for comparing different transformations in terms of standard, `BoN`, and `WoN` win rate.

In the first plot, we consider standard win rate. In this case, it is known that the IPO objective is win rate optimal. As can be seen, the logarithmic transformation (i.e., best-of-$N$ distillation) also achieves a nearly optimal win rate, which was already observed by Yang et al. (2024); Gui et al. (2024). All other transformations are sub-optimal for standard win rate.

Next, we consider Best-of-2 and Best-of-4 win rate. Here, additionally we include `bon_fp`, which is obtained by deriving the fixed point of Corollary 2. As can be seen, the identity transformation is no longer optimal. The best tradeoffs are given by `bon_fp`. We also observe that $\exp(5x)$ and $\exp(10x)$ are almost as good as `bon_fp` for Best-of-2 and Best-of-4, respectively. Moreover, the identity transformation and logarithmic transformation are sub-optimal in these cases, which

shows that considering standard win rate as the only metric is not optimal when inference-time procedure is concerned. We also observe that the behavior of identity transformation and logarithmic transformation is different in that the identity transformation gives better tradeoffs.

Finally, we consider Worst-of-2 and Worst-of-4 win rate. Again, it can be observed that won_fp gives the best tradeoffs for this inference-time procedure. Here, $\exp(-5x)$ and $\exp(-10x)$ are almost as good for Worst-of-2 and Worst-of-4, respectively. We also observe that identity transformation and logarithmic transformation are sub-optimal in these cases and the logarithmic transformation gives better tradeoffs for WoN compared to the identity transformation.

The above results demonstrate the importance of considering the inference-time procedure when performing alignment. We find that exponential transformation with different $t$'s are good for different inference-time procedures, which is our focus in practical experiments.

**Remark on additional compute**: While the calibration step induces extra computation overhead, we remark that it involves only forward pass on the model and only needs to be performed once per prompt before performing the policy optimization algorithm. In our experiments, for K=100 roll outs, we take training steps equivalent to 80 epochs for all datasets. Based on the 2:1 FLOPS ratio between back propagation and forward pass, the reward calibration step takes about 29% of the total training time. We hypothesize that the trade-off between this computational overhead and performance gain is task-specific and should be studied in future work.

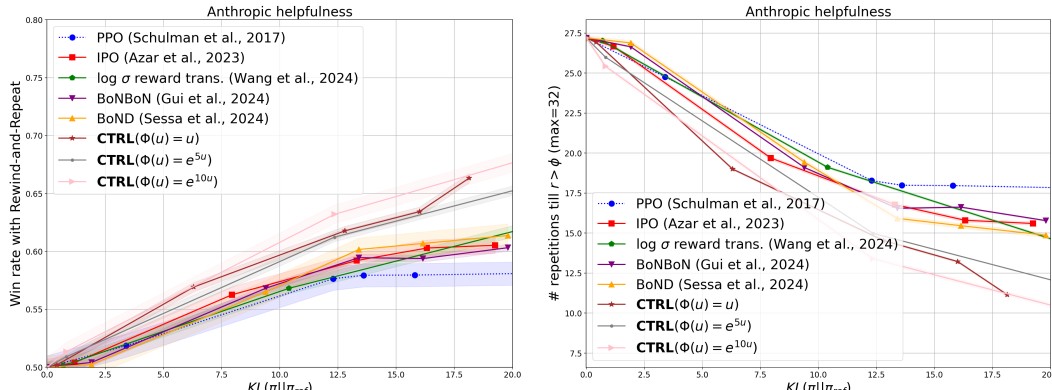

*Figure 12.* On the Rewind-and-Repeat inference-time strategy, we experiment with $\phi = 85\%$ile of the base policy's per-prompt reward on the Anthropic helpfulness dataset. Here, we see that exponential transformations continue to outperform baselines. In (a) we apply the rewind-and-repeat strategy (upto a maximum of 32 repeats), and then compare the win-rate of the aligned policy against the base policy. In (b), we measure the number of repeat-trials (maximum of 32 repeats) required to achieve a reward greater than the $\phi = 85\%$ile of the base policy.

## G. Analysis on Rewind-and-Repeat

We extended our study to an inference-time procedure, which is a variant of rejection sampling called *rewind-and-repeat*, motivated by recent work of (Beirami et al., 2024; Zhang et al., 2024). At inference-time, the procedure repeatedly generate independent samples from the policy until an outcome with a minimum reward threshold $\phi$ ($= 85\%$ile) is achieved or a pre-defined maximum number of generations $N(= 32)$ is reached. As shown in Fig 12, `InfAlign − CTRL` leads to improved performance in this adaptive-compute case as well.

