# OpenReview forum: "InfAlign: Inference-aware language model alignment"
_ICML.cc/2025/Conference — ICML 2025 poster_

### Official Review · Reviewer_XbH2 · 2025-02-14

**Overall Recommendation:** 4

**Summary:**

This paper explores a novel problem in LLM alignment **considering inference-time procedures**. More specifically, it aims to maximize the reward given a fixed LLM and an inference-time procedure, using reinforcement learning (RL). The focus is on Best-of-N as the inference-time procedure, while also providing general mathematical guidance for other potential inference-time procedures.

## update after rebuttal
I decide to keep my high score for this good paper. This paper should be accepted IMO!

**Claims And Evidence:**

Overall, the claims made in the submission are supported by clear and convincing evidence.

**Essential References Not Discussed:**

N/A

**Experimental Designs Or Analyses:**

The soundness/validity of experimental designs and analyses makes sense.

**Methods And Evaluation Criteria:**

The proposed method and evaluations are reasonable.

However, the evaluations seem somewhat limited. Demonstrating improvements in general instruction-following tasks would likely generate more excitement.

**Other Comments Or Suggestions:**

N/A

**Other Strengths And Weaknesses:**

**Weakness:** The paper focuses solely on Best-of-N as the inference-time procedure. While additional experiments may not be necessary, including the formulas for transformed rewards in other inference-time procedures could strengthen

**Questions For Authors:**

N/A

**Relation To Broader Scientific Literature:**

A key contribution of this paper is proposing a new problem setup, which is valuable. Studying how to improve a model during training while considering the inference-time procedures used for deployment is a meaningful direction.

**Theoretical Claims:**

I didn't check the theoretical claims, but there do exist a lot of proofs in the Appendix.

---

> ### Author Rebuttal · Authors · 2025-04-01
>
> We thank the reviewer for acknowledging the novelty and sufficiency of the provided evidence in our work. Below we address the reviewer’s concerns. We provide additional experimental results at https://drive.google.com/file/d/1pe4kZeu7JkW0e-o5QtQFwpnvkZ4xVI9s
>
> ***Q1: However, the evaluations seem somewhat limited. Demonstrating improvements in general instruction-following tasks would likely generate more excitement.***
>
> Thank the reviewer for the suggestion. We view demonstrating the method on more RLHF tasks as important next steps.
>
>
> ***Q2 Weakness: The paper focuses solely on Best-of-N as the inference-time procedure. While additional experiments may not be necessary, including the formulas for transformed rewards in other inference-time procedures could strengthen***
>
> Thanks for pointing this out. To demonstrate the generalizability of our framework, we consider another inference-time strategy called rewind-and-repeat, motivated by the recent work of [Beirami et al 2024, Zhang et al., 2024]. Given a pre-defined threshold $\phi$ on the reward, the procedure keeps generating responses until the response hits the threshold. In Figure 9 of the attached file, we show that our proposed method leads to better win-rate / KL tradeoff in these cases compared to other SOTA methods. We also show that the expected number of generations needed to achieve the predefined threshold is lower with our alignment method.
>
> Additional references (not listed in the paper):
> [Zhang et al., 2024] Yiming Zhang, Jianfeng Chi, Hailey Nguyen, Kartikeya Upasani, Daniel M. Bikel, Jason Weston, Eric Michael Smith. Backtracking Improves Generation Safety. Arxiv 2024.

---

> > ### Comment · Reviewer_XbH2 · 2025-04-01
> >
> > Thanks for your appreciation and more experiments. My concerns are addressed.
> >
> > Great work! I think this paper should be accepted.

---

### Official Review · Reviewer_dJSD · 2025-03-10

**Overall Recommendation:** 3

**Summary:**

This paper proposes the inference-aware alignment (InfAlign) framework to optimize model's inference-time win rate when various decoding strategies $\mathcal T$, e.g. Best-of-N, are applied.
The authors solve the KL-regularized win-rate maximization problem using an equivalent KL reward-maximization problem, where the objective reward $\mathcal R$ is calculated using the calibrated reward $\mathcal C_r$ and inference-time strategy $\mathcal T$.
Then they prove that for some inference strategies $\mathcal T$, which they denote as *"calibrated inference-time procedure"*, the reward objective can be derived as $\mathcal R_\Phi=\Phi(\mathcal C_r)$ using a transformation $\Phi$ independent of $r$ and $\pi_\mathrm{ref}$.
Such an ideal property applies for the BoN and WoN strategies, and the authors provide equations to characterize the solution of $\Phi$ for BoN/WoN, along with a practical exponential transforming function $\Phi_t$ to approximate the solution.
Finally, they provide an empirical estimation of $\mathcal C_r$ for practical implementation and utilize PPO to solve the KL-RL problem.
The empirical results show that: (1) the introduction of calibrated reward instead of original reward function improves model winrate; (2) the proposed method improves BoN decoding model's winrate.

**Claims And Evidence:**

The authors claim that training using calibrated reward improves model's winrate compared to the other reward optimization baselines (Sec. 5.3).
However, as depicted in Figure 4, CTRL's performance is quite close to that of BoND.

**Essential References Not Discussed:**

N/A

**Experimental Designs Or Analyses:**

The experiment setting in Sec. 5.2 is strange to me.
The calibrated reward measures the expected winning rate over SFT policy, and thus depends on the SFT model.
It's natural that the learned reward function differs from the calibrated reward.
The authors should discuss how the mismatch between learned reward and calibrated reward can hurt the performance.

**Methods And Evaluation Criteria:**

Unlike most existing work that optimizes for reward and use reward to assess model performance, this paper uses the win-rate over SFT model for training objective and evaluation.
However, there is a potential issue of reward hacking when relying solely on win rate over a fixed model.
To strengthen the evaluation, the authors could include comparisons of raw rewards and win rates over other models (e.g., models trained with different methods).

**Other Comments Or Suggestions:**

N/A

**Other Strengths And Weaknesses:**

Other Strengths:
- Theoretical soundness: this paper provides a clear and rigorous theoretical framework for inference-aware alignment, with well-justified derivations and proofs.

Other Weaknesses:
- The paper focuses primarily on BoN and WoN strategies. The generalization of the proposed framework to other inference-time strategies (e.g., self-consistency, chain-of-thought) is not discussed.

**Questions For Authors:**

How much additional computational overhead does the calibrated reward estimation introduce?

**Relation To Broader Scientific Literature:**

N/A

**Theoretical Claims:**

While I did not verify the detailed proofs, the derivations and arguments presented are clear and logically consistent.

---

> ### Author Rebuttal · Authors · 2025-04-01
>
> We thank the reviewer for acknowledging the theoretical soundness of our work. Below we address the reviewer’s concerns. We provide additional experimental results at https://drive.google.com/file/d/1pe4kZeu7JkW0e-o5QtQFwpnvkZ4xVI9s
>
> ***Q1: The authors claim that training using calibrated reward improves model's winrate compared to the other reward optimization baselines (Sec. 5.3). However, as depicted in Figure 4, CTRL's performance is quite close to that of BoND***
>
> We acknowledge that the tradeoff curve for standard win rate of our method is close to that of BoND, one of the SOTA methods (first row of Figure 4). However, for Anthropic helpfulness and harmlessness datasets, we do see a small improvement in the win-rate compared to BoND.
>
> Moreover, we would like to emphasize that the focus of the paper is on improving the inference-time win-rate, such as best-of-4 win rate or worst-of-4 win rate. We note that for these cases, the improvement we get over BoND is more pronounced, as shown in the second row of Figure 4. We mainly present the results for standard win rate as a side product of our investigation on the effect of reward calibration. We believe the fact that it is on par (and better on two of the three tasks) with SOTA methods is a convincing evidence.
>
>
> ***Q2: Unlike most existing work that optimizes for reward and use reward to assess model performance, this paper uses the win-rate over SFT model for training objective and evaluation. However, there is a potential issue of reward hacking when relying solely on win rate over a fixed model.  To strengthen the evaluation, the authors could include comparisons of raw rewards and win rates over other models (e.g., models trained with different methods).***
>
> Regarding using win-rate as the evaluation metric, see response to Q1 of reviewer QsSA. Further, we have evaluated the raw rewards of the models in Figure 10 of the attached file that show that the raw rewards are correlated with the win-rates shown in Figure 4 (top left).
>
> ***Q3: The calibrated reward measures the expected winning rate over SFT policy, and thus depends on the SFT model. It's natural that the learned reward function differs from the calibrated reward. The authors should discuss how the mismatch between learned reward and calibrated reward can hurt the performance.***
>
> The review is correct that the calibrated reward might differ from the learned reward function. However, the reward models are usually learned from human preference data using a Bradley Terry model to be used as proxies for real human performance. Hence, the expected win rate is a more robust metric compared to the learned reward model. For more discussions on the use of  win-rate as the evaluation metric, see response to Q1 for reviewer QsSA.
>
> In this paper, we show that instead of hurting, the calibrated reward consistently improves the standard win rate as shown in the first row of Figure 4, where CTRL with $\Phi(u) = u$ represents PPO with calibrated reward and PPO represents PPO with raw reward.
>
> *(Calibration in fact mitigates reward hacking)* Moreover, we conduct controlled experiments to show that calibration in fact mitigates reward hacking, as discussed in Appendix D. To induce reward hacking, we injected specific phrases into the start of preferred responses of our preference datasets: “Sorry, I can’t help you with that” for Harmlessness and “Sure” for Helpfulness. We then evaluated the model’s accuracy on a poisoned evaluation set where these phrases were inverted (added to the unpreferred responses). A significant drop in accuracy on this poisoned set would indicate reward hacking: a reliance on the spurious correlation. Figure 5 in the appendix shows that calibrated reward models are more robust to these manipulated correlations, maintaining higher accuracy compared to uncalibrated models. Thus, calibration improves the reward model’s robustness to hacking based on training data poisoning.
>
>
> ***Q4: The generalization of the proposed framework to other inference-time strategies (e.g., self-consistency, chain-of-thought) is not discussed.***
>
> In Figure 9 of the attached file, we consider an additional inference-time strategy called Rewind-and-Repeat inference-time strategy on the Anthropic helpfulness task. We refer to the response to Q2 of Reviewer XbH2.
>
> ***Q5: How much additional computational overhead does the calibrated reward estimation introduce?***
>
> While the calibration step induces extra computation overhead, we remark that it involves only forward pass on the model and only needs to be performed once per prompt before performing the policy optimization algorithm. In our experiments, we take training steps equivalent to 80 epochs for all datasets. Based on the 2:1 FLOPS ratio between back propagation and forward pass, the reward calibration step takes about 29% of the total training FLOPS.

---

> > ### Comment · Reviewer_dJSD · 2025-04-03
> >
> > Thanks for your effort and reply. I will maintain my score. Best of luck with your work.

---

### Official Review · Reviewer_QsSA · 2025-03-11

**Overall Recommendation:** 4

**Summary:**

The paper proposes a new alignment method based on RL to optimize the Best-of-N and Worst-of-N performance of language models. They define the alignment problem as optimizing the win rate against the reference policy minus the KL penalty. To solve the alignment problem under some inference-time procedure, they use calibrated reward (win rate of the response against the reference policy) after some transformation. They show that for BoN and WoN sampling, the optimal win rate and kl divergence is independent of the reward and the reference policy and the exponential transformation can approximate the optimal transformation for win rate and kl divergence. Empirical results show that the alignment method is competitive for standard win rate and superior for BoN and WoN win rate.

## update after rebuttal
The authors' rebuttal addresses my concern well. Therefore I decided to raise my score to 4.

**Claims And Evidence:**

Yes.

**Essential References Not Discussed:**

No

**Experimental Designs Or Analyses:**

Yes. I think the experimental results are valid for the defined problem.

**Methods And Evaluation Criteria:**

I am skeptical of the problem setting. The reward in definition 5 is the win rate against the reference policy. Obviously this is different from the original RLHF problem. And I am not sure it is always reasonable, especially when the reference policy is weak. The experiments also report the win rate against the reference policy.

**Other Comments Or Suggestions:**

No.

**Other Strengths And Weaknesses:**

No.

**Questions For Authors:**

1. What is the accuracy improvement of the larger reward model based on PaLM-2 M against PaLM-2 S? Is it enough to model the generalization error of reward models in practice?
2. When measuring the BoN win rate in the experiments, do you use true rewards or learned rewards for sampling?

**Relation To Broader Scientific Literature:**

Previous alignment methods often assume direct sampling from language models during inference. In practice, more complex sampling methods like BoN sampling might be used. The paper proposes a method to optimize the performance of BoN (or WoN) sampling to overcome the drawback.

**Theoretical Claims:**

No

---

> ### Author Rebuttal · Authors · 2025-04-01
>
> We thank the reviewer for the detailed reading and comments. Below we address the reviewer’s questions in detail. We provide additional experimental results at https://drive.google.com/file/d/1pe4kZeu7JkW0e-o5QtQFwpnvkZ4xVI9s
>
> ***Q1: I am skeptical of the problem setting. The reward in definition 5 is the win rate against the reference policy. Obviously this is different from the original RLHF problem. And I am not sure it is always reasonable, especially when the reference policy is weak. The experiments also report the win rate against the reference policy.***
>
> Rooted in preference learning, reporting the win-rate vs KL tradeoff is a standard practice in the RLHF literature to evaluate the effectiveness of the language model alignment, ranging from the canonical work of Stiennon et al 2020 and the blog of Hilton and Gao 2022, which use win-rate with human preferences, to more recent works of Eisenstein et al., 2024; Mudgal et al., 2024, which uses more powerful models as judges. These win rates are direct reflections of human preferences while alternatives such as expected reward scores are indirect proxies, especially in cases where these reward models are learned from human preference data using a Bradley Terry model and their raw values may not have a physical meaning.
>
> Moreover, Azar et al 2023, Gui et al 2024 formally formulated the optimization for win-rate vs KL tradeoff as the objective of RLHF. We follow these works and generalize the objective to include inference-time win rates, which better suits the modern regime of increasing inference-time compute. Hence we believe considering the win rate as the RLHF objective should not be viewed as a limitation of the work.
>
> Further, we have evaluated the raw rewards of the models in Figure 10 of the attached file, which shows that the raw rewards are correlated with the win-rates shown in Figure 4 (top left).
>
> Additional references (not listed in the paper):
> [Hilton and Gao 2022]  Hilton, J. and Gao, L. Measuring Goodhart’s law, April 2022. URL https://openai.com/research/measuring-goodharts-law. Accessed: 2024-0103.
>
> ***Q2: What is the accuracy improvement of the larger reward model based on PaLM-2 M against PaLM-2 S? Is it enough to model the generalization error of reward models in practice?***
>
> For Anthropic helpfulness dataset, the pairwise preference accuracy increases from 73.0% to 77.7%. There is still a gap between the accuracy of the large model and the true human preference, which is indeed a limitation of our evaluation approach. However, we want to remark here that collecting real human preference data is costly, and prior work [Stiennon et al., 2022, Wang et al., 2024] takes a similar approach where a larger and more accurate reward model is used as a judge to compute win-rate over models aligned with smaller reward models. We will add discussions on this limitation in the revised draft.
>
> ***Q3: When measuring the BoN win rate in the experiments, do you use true rewards or learned rewards for sampling?***
>
> Following the literature (e.g., Eisenstein et al., 2024; Mudgal et al., 2024), we use a separate, more powerful model than the reward model as the judge to measure the win rate. More specifically, we use the PaLM-S fine-tuned reward model during RL training and BoN/WoN selection. We then evaluate the inference-time win-rate w.r.t base reference policy model using the PaLM-M reward model.

---

> > ### Comment · Reviewer_QsSA · 2025-04-02
> >
> > I think the accuracy gap between PaLM-S and PaLM-M (73.0% vs 77.7%) is not large enough to represent the gap between the accuracy of a reward and human preference (maybe 77.7% vs 100%). This reduces the reliability of the experiment results, including the BoN win rate in Q3.
> >
> > I think the answer to Q1 addresses my concern well. Above all, I decide to keep my score.

---

> > > ### Author Response · Authors · 2025-04-02
> > >
> > > We thank the reviewer for the prompt response and acknowledging our response to Q1. Regarding the gap between model judgement and human preference, we want to note that the helpfulness and harmlessness dialog, and text summarization tasks are subjective. In fact, prior work [Sec 3.3 in Stiennon et al 2020, Sec 3.4.1 in Bai et al 2022] has noted an inter-rater agreement often less than 77% (expert-non-expert agreement is even less). Hence we should expect no better accuracy even with LLM-judges fine-tuned on such data. While we do acknowledge these limitations, it is an issue that is shared in a large body of work that uses models trained on preference data where there is interannotator disagreement. Developing more reliable ways for this is beyond the scope of this work. RewardBench [Lambert et al., 2024], a reward model leaderboard (https://huggingface.co/spaces/allenai/reward-bench) currently has reward model accuracy upper bounded by 75.7%, 72.3%, and 76.7% on the Anthropic Helpfulness, Harmlessness, and Reddit text summarization preference datasets respectively, and hence our evaluation model accuracies (77.7%, 77.0, and 76.4%) are in fact on par with SOTA performance on these datasets. We hope this addresses the reviewer’s concern. We will add more discussions to this in the revised version.
> > >
> > > [Additional references] (not in the paper)
> > > [Lambert et al., 2024] Nathan Lambert, Valentina Pyatkin, Jacob Morrison, LJ Miranda, Bill Yuchen Lin, Khyathi Chandu, Nouha Dziri, Sachin Kumar, Tom Zick, Yejin Choi, Noah A. Smith, Hannaneh Hajishirzi. RewardBench: Evaluating Reward Models for Language Modeling. 2024

---

### Official Review · Reviewer_6Tis · 2025-03-13

**Overall Recommendation:** 3

**Summary:**

The paper introduces a new problem called InfAlign and proposes a method called InfAlign-CTRL to solve it. Their theoretical properties are investigated. Their main claims are (1) InfAlign-CTRL with no inference procedure improves the standard win rate due to the reward calibration, and (2) InfAlign-CTRL with BoN/WoN improves the win rate after the inference-time procedures.

## update after rebuttal

Overall, each theoretical/experimental result (including the rebuttal) seems convincing to some extent, but there still remain concerns about the intrinsic dependence on N and the heuristic derivation of the actual transformation used in experiments. So I will keep the score as is.

**Claims And Evidence:**

**Claims:**
- The paper first introduces the notion of calibrated rewards as the win rate in terms of a given reward.
- Based on it, the paper formulates a new problem called InfAlign, which is the maximization of win rate after some inference-time procedure like BoN/WoN, and derives a reformulation as the standard reward maximization problem with some transformed calibrated reward.
- Also they propose a method called InfAlign-CTRL to solve InfAlign with reward calibration and transformation.
- Their main claims are (1) InfAlign-CTRL with no inference procedure improves the standard win rate due to the reward calibration, and (2) InfAlign-CTRL with BoN/WoN improves the win rate after the inference-time procedures.

**Evidences/Derivations:**
- The derivation of InfAlign is straightforward given the motivation that we want to directly optimize the performance of inference-time alignments.
- Its reformulation as the (transformed) reward maximization is derived under the assumption that both the win-rate and inference-time procedure are differentiable, which seems unrealistic for BoN/WoN methods.
- The actual transformations used in this paper for BoN/WoN are heuristically derived in Section 4 as exponential transformations with a hyperparameter t dependent on N.
- Experiments are supposed to validate the main claims. However, (1) it seems not sufficiently backed why the maximization of calibrated rewards leads to improved and robust alignments, and (2) Figure 4 actually shows the improvement in BoN/WoN with the proposed method, but the most of results are shown with only N=4 and there are no argument on the possibility of overfitting to N. Intuitively, the proposed method seems to require the hyperparameter search on t and retraining of the aligned model for different N's.

**Essential References Not Discussed:**

N/A

**Experimental Designs Or Analyses:**

See Methods And Evaluation Criteria.

**Methods And Evaluation Criteria:**

The experimental setups and evaluation protocols overall make sense. However, I'm concerned that (i) whether or not the PaLM-2 M model is appropriate for true rewards of the given datasets, which could affect the reward calibration and its analyses, and (ii) how the InfAlign-CTRL trained with fixed N performs with various N's in test time, which is crucial for practical applications of inference-time procedures like BoN/WoN.

**Other Comments Or Suggestions:**

N/A

**Other Strengths And Weaknesses:**

Major strengths: (1) novelty of the ideas of calibrated rewards and direct optimization of win rate after inference-time alignment; (2) theoretical investigations on the InfAlign problem.

Major weaknesses: (1) seemingly unrealistic assumptions in theoretical results; (2) the heuristic derivation of exponential transformations; (3) the possibility of overfitting to the fixed N used in training.

**Questions For Authors:**

See Claims And Evidence.

**Relation To Broader Scientific Literature:**

The paper proposed a general framework that can be applied to various inference-time procedures such as Best-of-N (BoN) and its variants, to improve their peformance in terms of win rates. Also they introduced a novel notion called calibrated rewards, which itself is of independent interest.

**Theoretical Claims:**

The theoretical claims seem valid but some assumptions would not be satisfied in the real world, as briefly stated in Claims and Evidences.

---

> ### Author Rebuttal · Authors · 2025-04-01
>
> We thank the reviewer for acknowledging the novelty of proposed method and theoretical investigations. Below we address the reviewer’s concerns and provide additional experimental results: https://drive.google.com/file/d/1pe4kZeu7JkW0e-o5QtQFwpnvkZ4xVI9s
>
> ***Q1: …the assumption that both the win-rate and inference-time procedure are differentiable … seems unrealistic for BoN/WoN …***
>
> We acknowledge that Theorem 1 for general inference-time procedures needs the assumption on differentiability of the inference-time win-rate. However, this assumption naturally holds for the considered procedures of BoN/WoN.
>
> To see this, under a fixed prompt $\mathbf{x}$, each policy $\pi(\cdot \mid \mathbf{x})$ can be viewed as a $\mathcal{Y}$-dimensional vector. For a pair of policies $\pi\_1$ and $\pi\_2$, the win-rate as defined in Definition 3 is an expectation of the win random variable with samples from $\pi\_1(\cdot \mid \mathbf{x})$ and $\pi\_2(\cdot \mid \mathbf{x})$. Hence it is a linear function of both policies, making it differentiable with respect to both policies. Correspondingly, the inference-time win rate as defined in Definition 4 is linear with respect to both inference-time policies. By the chain rule, it is sufficient to show that the inference-time policy is differentiable. For BoN/WoN, the inference-time policy has explicit forms stated in Lemma 5. For example, $BoN\_{\pi}(\mathbf{y} \mid \mathbf{x}) = N pi(\mathbf{y} \mid \mathbf{x}) \mathcal{C}\_{r, \pi} (\mathbf{x}, \mathbf{y})^{N\_1}$. Since $\mathcal{C}\_{r, \pi} (\mathbf{x}, \mathbf{y})$ is a linear function of $\pi(\cdot \mid \mathbf{x})$ as defined in Definition 2, for all $\mathbf{y}$, $BoN\_{\pi}(\mathbf{y} \mid \mathbf{x})$ is a polynomial function of the base policy $\pi(\cdot \mid \mathbf{x})$, making it differentiable w.r.t. $\pi(\cdot \mid \mathbf{x})$. These two facts combined leads to differentiability of inference-time win-rate for BoN/WoN. We will make the above discussion clear in the updated version.
>
> ***Q2: The actual transformations for BoN/WoN are exponential transformations with a hyperparameter t dependent on N. …. requires the hyperparameter search on t and retraining of the aligned model for different N's.***
>
> While we use the exponential transformation in the empirical experiments, we did show that they lead to near optimal KL / win-rate tradeoff curves in Section 4.3. For example, in Figure 2 (a), the curve marked bon_fp is obtained by using the transformations obtained from solving the fixed point in Corollary 2. As shown, the tradeoff curve is numerically close compared to the one obtained from $exp(10u)$. Additionally, the exponential transformation is motivated by exponential tilting of loss functions (Li et al., 2021; 2023), which has been shown to help optimize different quantiles of the reward with different values of t, making suitable transformation for the BoN/WoN inference-time procedure.
>
> When choosing the hyperparameter $t$, our method doesn’t need to retrain different models to perform the hyperparameter search. Instead, the search can be done efficiently using analytical tools with closed form expressions on KL and win rate (Theorem 3). To demonstrate that the findings will generalize to practical settings, we present more results in the attached file. For the analytical analysis (middle column of Figure 7 in the submission), $e^{5u}$ works better than $e^{10u}$ for best-of-2 and the reverse holds for best-of-4. In Figure 8 of the attached file, we show that for real models and tasks, we see the same trend, demonstrating the transferability.
>
>
> ***Q3: Figure 4 .. most of results are shown with only N=4 and there are no argument on the possibility of overfitting to N.***
>
> Regarding the overfitting to $N$, we do find different $t$’s work the best for different $N$s. However, we would like to mention that in many practical settings, the $N$ that is going to be used at inference-time is known during the RLHF phase. And we could find the best exponent $t$ efficiently for the $N$ to be used as mentioned in the response to Q2. In the attached file, we also provide results for N=2 and N=32 to demonstrate the generality of our approach.
> In cases where $N$ might change due to the deployment resources, we show that while the best $t$ is different for different $N$’s, the gains are generalizable for mismatched cases as well.  In Figure 8 of the attached file, we obtain consistent gains for different $N$s with different $t$’s. For example, with $t = 10$, we obtain significant gains for all three cases of $N = 2, 4, 32$, showing that overfitting to $N$ is not a major limitation.
>
> ***Q4: why the maximization of calibrated rewards leads to improved and robust alignments***
>
> Response: In Appendix D, we present experiments to demonstrate that calibrated reward models are less susceptible to reward hacking. See response to Q3 of reviewer dJSD for more discussion on this. We will add more discussions in the future revisions.

---

> > ### Comment · Reviewer_6Tis · 2025-04-03
> >
> > > We acknowledge that Theorem 1 for general inference-time procedures needs the assumption on differentiability of the inference-time win-rate. However, this assumption naturally holds for the considered procedures of BoN/WoN. (...)
> >
> > Thank you for the clarification. I've been convinced about this.
> >
> > However, the rebuttal does not resolve my concern that the exponential transformations are heuristic and its derivation is just motivated by the previous literature. Also my concern about the inherent dependence of the trained model on N has not been resolved.
> >
> > > However, we would like to mention that in many practical settings, the N that is going to be used at inference-time is known during the RLHF phase.
> >
> > I disagree this. I think one of the major advantages of inference-time alignment is the flexibility of the choice of N, i.e., we can easily control the tradeoff between accuracy and computational budgets.
> >
> > Note: I could not access the provided URL.

---

> > > ### Author Response · Authors · 2025-04-03
> > >
> > > >"Note: I could not access the provided URL"
> > >
> > > We apologize for the nonfunctional link. We believe both of the following links will work now (including both to be safe.): (1) https://github.com/infalign/infalign/blob/999686b3305a93992ad45a716f56c64ed1ffe177/InfAlign-rebuttal.pdf; (2) https://drive.google.com/file/d/1pe4kZeu7JkW0e-o5QtQFwpnvkZ4xVI9s.
> > >
> > > >"the inherent dependence of the trained model on N has not been resolved."
> > >
> > > To show that our method can adapt to the case where $N$ may not be known ahead, we provide additional experiments to show that while the best $t$ is different for different $N$’s, the gains are generalizable for mismatched cases as well.  In Figure 8 of the attached file, we obtain consistent gains for different $N$s with different $t$’s. For example, with $t = 10$, we obtain significant gains for all three cases of $N = 2, 4, 32$, showing that overfitting to $N$ is not a major limitation.
> > >
> > > >"I disagree this (N that is going to be used at inference-time is known during the RLHF phase). I think one of the major advantages of inference-time alignment is the flexibility of the choice of N, i.e., we can easily control the tradeoff between accuracy and computational budgets.”
> > >
> > > We would like to emphasize that the inference logic for large-scale LLM serving needs to be chosen and fixed in advance of deployment, and hence the model could be specifically finetuned for the chosen inference-time procedure. We do agree with the reviewer that the inference-time procedure might be more complex than standard best-of-N in some practical scenarios where the number of trials N, may be chosen based on a variety of factors to provide a good scaling behavior. We would also like to mention that we have extended our study to a variant of rejection sampling with variable N such that the trials are rewinded and repeated until an outcome with a minimum reward threshold is achieved. And we have shown that InfAlign-CTRL leads to improved performance in this adaptive-compute case as well. Results are in Figure 9 of the attached URL. And we provide a more detailed description of the procedure in the response to Q2 of Reviewer XbH2.
> > >
> > > >"rebuttal does not resolve my concern that the exponential transformations are heuristic and its derivation is just motivated by the previous literature."
> > >
> > > (1) We solved the optimal transformation analytically and observed that the exponential transformation is almost optimal. So, while we were inspired by previous literature to try out and design this transformation as a good option, it is actually almost optimal within an additive 1% of optimal win rate at any KL divergence (see Figure 2 and Figure 7). Thus, the transformation actually comes with a near-optimal guarantee in inference-time win rate even though we agree that the design is heuristic.
> > >
> > > (2) We agree that it is also important to compare the empirical gap between our heuristic and optimal transformations. We will train a new model with the learned fixed point transformation. In practice, this can be done by storing the learned transformation via a lookup table of size K (=100) that is fixed for all prompts. Per our theoretical results (see Corollary 2 in Sec 4.3), we don't expect the tradeoff curves to be different from those of the almost optimal exponential transformations. Training and evaluating the new models will take a few days and we will include them in the github repositorywhen we get them: https://github.com/infalign/infalign/blob/999686b3305a93992ad45a716f56c64ed1ffe177/InfAlign-rebuttal.pdf.

---

### Decision · Program_Chairs · 2025-05-01

**Decision:**

Accept (poster)

**Comment:**

This paper introduces InfAlign, a novel RLHF framework designed to optimize language model performance for specific inference-time procedures like Best-of-N (BoN). It addresses the train/test mismatch in standard RLHF by maximizing the inference-time win rate against a base model. This is achieved by optimizing a KL-regularized objective using a transformed version of a "calibrated reward" (essentially the win rate). The authors show theoretically and empirically that this approach, particularly with a near-optimal exponential transformation for BoN/WoN, improves inference-time win rates (3-8% reported for BoN). Reward calibration alone is also presented as a strong baseline improving standard win rates and robustness.

Reviewers acknowledged the paper's novelty in addressing inference-aware alignment and introducing calibrated rewards. The theoretical framework was generally seen as sound. Key strengths included the novel problem formulation, theoretical grounding, and empirical gains in inference-time win rates. Despite minor concerns about the practical implications of the heuristic transformation and the method's dependence on a pre-defined N temper enthusiasm, the authors’ rebuttal addressed these concerns. Based on the solid contribution, I recommend to accept this paper.